# Dissecting Misalignment of Multimodal Large Language Models via Influence Function

## Abstract

Multi-modal Large Language models (MLLMs) are always trained on data from diverse and unreliable sources, which may contain misaligned or mislabeled text-image pairs. This frequently causes robustness issues and hallucinations, leading to performance degradation. Data valuation is an efficient way to detect and trace these misalignments. Nevertheless, existing methods are computationally expensive for MLLMs. While computationally efficient, the classical influence functions are inadequate for contrastive learning models because they were originally designed for pointwise loss. Additionally, contrastive learning involves minimizing the distance between the modalities of positive samples and maximizing the distance between the modalities of negative samples. This requires us to evaluate the influence of samples from both perspectives. To tackle these challenges, we introduce the Extended Influence Function for Contrastive Loss (ECIF), an influence function crafted for contrastive loss. ECIF considers both positive and negative samples and provides a closed-form approximation of contrastive learning models, eliminating the need for retraining. Building upon ECIF, we develop a series of algorithms for data evaluation in MLLM, misalignment detection, and misprediction trace-back tasks. Experimental results demonstrate our ECIF advances the transparency and interpretability of MLLMs by offering a more accurate assessment of data impact and model alignment compared to traditional baseline methods.

## 1 Introduction

Multi-modal Large Language models (MLLMs) (Yin et al., 2023; Koh et al., 2024) have garnered significant attention for their ability to integrate and understand various data types, such as image, text, and audio. Despite their growing application, existing MLLMs often suffer from robustness issues (Carlini & Terzis, 2021) and hallucinations, primarily stemming from misaligned text-image pairs in the training data (Kim et al., 2023). These misalignments, manifesting as semantic mismatches, contextual inconsistencies, or discrepancies between abstract and concrete elements, can severely degrade model performance. MLLMs assume consistent alignment between image-text pairs, but when this assumption fails, it leads to incorrect interpretations, ultimately degrading model performance. Consequently, improving dataset transparency is crucial, as model developers need the ability to trace and identify problematic data samples. However, diagnosing issues caused by misaligned data, such as mislabeled or biased samples, is difficult when working with large text-image datasets.

Although the critical role of training data in shaping MLLM capabilities is well recognized, there remains a lack of robust evaluation mechanisms for data quality (Nguyen et al., 2022). To address this, various data valuation methods (Jia et al., 2019; Ghorbani & Zou, 2019; Yoon et al., 2020; Han et al., 2020) have been introduced to enhance dataset transparency by quantifying the contribution of individual data points to model performance. These approaches typically assign higher contribution scores to training instances whose inclusion significantly boosts model performance compared to their exclusion. Some methods, such as Shapley Value (Kwon & Zou, 2022), require multiple retraining processes with different subsets of data, which is computationally expensive and impractical for large models. To overcome this limitation, influence function-based methods have

gained popularity, as they estimate data contributions using gradient information, thereby avoiding retraining (Choe et al., 2024).

However, applying influence functions to MLLMs poses significant challenges. (i) First, the influence function was initially designed for M-estimators (Huber, 1981), which operate with pointwise loss. However, MLLMs rely on noise-contrastive estimation (Radford et al., 2021; Gutmann & Hyvärinen, 2010; He et al., 2020) as their training objective. This objective encourages the model to draw positive pairs closer in feature space while pushing negative pairs apart, making the influence function unsuitable for direct application to contrastive loss. (ii) Second, the influence of negative pairs in contrastive learning has gained increasing attention recently (van den Oord et al., 2019; Yuksekgonul et al., 2023). Robinson et al. (2021) emphasized the importance of negative samples, especially "hard" negatives - samples that are mapped close in feature space but should ideally be far apart. However, the original definition of the influence function does not consider the roles of positive and negative samples. This oversimplified analysis is particularly prone to underestimating the impact of certain hard negative samples on the learning process (Chen et al., 2020a). (iii) Lastly, calculating the necessary gradients and Hessian matrices for influence functions demands significant computational and memory resources, which becomes infeasible in the large-scale, high-dimensional context of MLLMs (Li et al., 2023a;b).

To address these challenges, we propose the *Extended Influence Function for Contrastive Loss (ECIF)*, a novel method designed to quantify data importance specifically for contrastive learning. ECIF enjoys a closed-form approximation of the original contrastive loss, thus eliminating the need for re-training - a process that is impractical in the era of large models. It also accounts for the dual role of data points as both positive and negative samples, providing a more comprehensive understanding of their impact on model training. This approach provides a more accurate measurement of misalignment. Our contributions are summarized as follows:

- We propose ECIF, the first dual-perspective data valuation method for MLLMs, which quantifies the impact of data points as both positive and negative samples. This comprehensive approach enables a more accurate measurement of data contribution, particularly addressing the influence of negative samples in contrastive learning.
- Based on ECIF, we develop corresponding algorithms for different tasks, including identifying the most valuable data (related to specific tasks), misalignment detection, and misprediction trace-back.
- Comprehensive experimental results demonstrate that ECIF can effectively and efficiently remove the influence of samples compared to retraining and identify influential data in the training set. Moreover, our methods based on ECIF are also effective in identifying influential data (harmful data and valuable data) for fine-tuning, mispredictions trace back, and detecting misaligned data.

## 2 RELATED WORK

**Contrastive Learning.** Recently, self-supervised contrastive learning (Chen et al., 2020b) has emerged as a highly effective approach for acquiring representations without the need for labeled data (Donahue & Simonyan, 2019). This model utilizes a contrastive loss, which pushes dissimilar data pairs apart while pulling similar pairs closer together. Contrastive learning plays a pivotal role in advancing MLLMs by integrating and understanding information across diverse modalities, such as text and images (Radford et al., 2021; Jiang et al., 2024). In multi-modal contrastive learning tasks, proper alignment of the training data ensures accurate cross-modal associations, enabling models to learn and extract consistent feature representations (Wang & Isola, 2020). One of the key challenges in training with noisy, large-scale image-text pairs sourced from the internet is achieving effective alignment between these modalities. To address this, researchers have developed various methods, such as those proposed by Gao et al. (2022) and Yao et al. (2021), which introduce finer-grained and more extensive interactions between text and images to improve cross-modal alignment. Despite extensive research on contrastive learning, we are the first to explore the interactive influence between pairs using influence functions. Our work bridges this gap by applying influence functions in contrastive learning, allowing for a deeper understanding of both positive and negative samples. This comprehensive approach enhances the accuracy of misalignment measurements in data pairs, providing a more thorough assessment of data valuation.

**Influence Function.** The influence function, initially a staple in robust statistics (Cook, 2000; Cook & Weisberg, 1980), has seen extensive adoption within deep learning since (Koh & Liang, 2017). Its versatility spans various applications, including detecting mislabeled data, interpreting models, addressing model bias, and facilitating machine unlearning tasks. For data removal, recent work using influence function including unlearning features and labels (Warnecke et al., 2023), forgetting a subset of image data for training deep neural networks (Golatkar et al., 2020; 2021), removing the influence of nodes and edges in graph neural networks Wu et al. (2023), and model debiasing (Chen et al., 2024). Besides, various studies have applied influence functions to interpret models across different domains, including natural language processing (Han et al., 2020) and image classification (Basu et al., 2021), while also addressing biases in classification models (Wang et al., 2019), word embeddings (Brunet et al., 2019), and finetuned models (Chen et al., 2020a). Recent advancements, such as the LiSSA method (Agarwal et al., 2017; Kwon et al., 2023; Grosse et al., 2023) and kNN-based techniques (Guo et al., 2021), have been proposed to enhance the computational efficiency of computing the influence function. Despite numerous studies on influence functions, we are the first to extend them to contrastive learning. Moreover, compared to traditional models, contrastive learning introduces additional complexity in influence function analysis, as it requires considering data points in both positive and negative roles. Our dual-perspective approach of ECIF offers a more comprehensive view of data impact, leading to more accurate measurements of misalignment in text-image pairs. Bridging the theoretical gap between positive and negative pairs has posed significant challenges in our work, which has been addressed in our proof.

## 3 PRELIMINARIES

**Contrastive Loss.** Contrastive loss is an effective tool in multi-modal models for aligning and learning relationships between different types of data, such as images and text [1]. Specifically, given a set of paired data consisting of text $x^T$ and image $x^I$, we aim to construct embedding vectors $u$ and $v$ for text and image respectively via the encoder parameterized as $\theta$. In a batch of $N$ text-image pairs, each pair $(x_k^T, x_k^I)$ is embedded as $(u_k, v_k)$. We denote the text embeddings for this batch as $U = (u_1, \ldots, u_N)$, and similarly, the image embeddings as $V = (v_1, \ldots, v_N)$.

The contrastive loss is designed to minimize the distance between embeddings of matching pairs while maximizing the distance between non-matching pairs. Define the cosine similarity function as $s(u, v) = \frac{u \cdot v^{\mathrm{T}}}{\|u\|\|v\|}/\tau$, where $\tau$ is a trainable temperature parameter. For brevity, we will omit detailing $\tau$ in subsequent discussions. For each batch, we construct a similarity matrix $S$ with $S_{i,j} = s(u_i, v_j)$. Then, the self-supervised contrastive loss is defined as

$$L_{\text{Batch}}(U, V; \theta) = \sum_{i=1}^{N} -\log(e_i \cdot \sigma(S_{i,*}) - \log(e_i \cdot \sigma(S_{*,i}^{\mathrm{T}}))) \tag{1}$$

$$= \sum_{i=1}^{N} L_{T2I}(u_i, V; \theta) + L_{I2T}(v_i, U; \theta), \tag{2}$$

where $e_i$ is the $i$-th standard basis vector in $N$-dimensional space, $\sigma$ is softmax function. Observing from (1), we can separate the loss to image-to-text (I2T) and text-to-image (T2I) denoted in (2) and define loss function on similarity matrix as $L_{T2I}(S; \theta)$ (and $L_{I2T}(S; \theta)$). We will incorporate an $L_2$ regularization term into the loss function, which allows us to avoid overfitting. Thus, for a given set of batches $\mathcal{B}$, the objective loss can be written as

$$L_{\text{Total}}(\mathcal{B}; \theta) = \sum_{(U,V) \in \mathcal{B}} L_{\text{Batch}}(U, V; \theta) + \frac{\delta}{2} \|\theta\|_2^2. \tag{3}$$

**Influence Functions.** The influence function quantifies how an estimator relies on the value of each individual point in the sample. Consider a neural network $\hat{\theta} = \arg\min \sum_{i=1}^{n} \ell(z_i; \theta)$ with pointwise loss function $\ell$ and dataset $D = \{z_i\}_{i=1}^{n}$. When we remove a point $z_m$ from the training dataset, the corresponding optimal model is denoted as $\hat{\theta}_{-z_m}$. The influence function provides an

---

[1]For simplicity, we focus on two modalities (text and image) in the paper. Our method can be generalized to multi-modalities directly.

efficient way to approximate $\hat{\theta}_{-z_m} - \hat{\theta}$ for a strongly convex and twice differentiable $\ell$. By up-weighing $z_m$ by $\epsilon$, we denote the substitutional parameter via the response function as

$$\hat{\theta}_{-z_m}(\epsilon) = \arg\min \frac{1}{n} \sum_{i=1}^{n} \ell(z_i; \theta) + \frac{\epsilon}{n} \cdot \ell(z_m; \theta). \tag{4}$$

Then we can obtain an estimator for the actual change in parameters as:$\lim_{\epsilon \to -1} \hat{\theta}_{-z_m}(\epsilon) - \hat{\theta} = -H_{\hat{\theta}}^{-1} \cdot \nabla_\theta \ell(z_m; \hat{\theta})$, where $H_{\hat{\theta}} = \sum_{i=1}^{n} \nabla_\theta^2 \ell(z_i; \hat{\theta}) + \delta I$ is the Hessian matrix at the point of $\hat{\theta}$.

For a differentiable model evaluation function ff, such as calculating the total model loss over a test set, the change resulting from removing $z_m$ in the evaluation results can be approximated by

$$f(\hat{\theta}_{-z_m}) - f(\hat{\theta}) \approx \nabla_\theta f(\hat{\theta})(\hat{\theta}_{-z_m} - \hat{\theta}) \approx -\nabla_\theta f(\hat{\theta}) \cdot H_{\hat{\theta}}^{-1} \nabla_\theta \ell(z_m; \hat{\theta}).$$

Scaling gradient-based methods to MLLMs is challenged by the high computational and memory demands due to the gradients' high dimensionality. Choe et al. (2024) introduced a low-rank gradient projection algorithm (LOGRA) to enhance the efficiency of gradient projection. They observed that the gradient from backpropagation is structured as a sum of Kronecker products of forward and backward activations. LOGRA applies an additional Kronecker-product structure to the projection matrix $P \triangleq P_i \otimes P_o$. It first projects the forward and backward activations onto low-dimensional spaces using $P_i$ and $P_o$, respectively, and then reconstructs the projected gradient directly from these reduced activations. For more details, see Appendix B.

## 4  INFLUENCE FUNCTION IN CONTRASTIVE LEARNING

In this section, we will consider how to estimate the value of a given sample $(x^T, x^I)$ in the contrastive loss (3) using the influence function method. Generally, in the original influence function method, a term in the loss function which only contain the fully information from the target sample is up-weighted by $\epsilon$. Then, a response function in (4) related to $\epsilon$ is derived. Within this analytical framework, when $\epsilon$ is set to $-1$, the resultant loss and model parameters are the same as those obtained by removing the sample via retraining. However, in the context of contrastive learning, because the information of the sample point appears in every term of the loss function for its batch, it is not feasible to isolate the relevant information of this sample within a batch into an independent term and then perform an up-weight operation on this sample to derive the influence function.

Thus, we need to execute fine-grained analysis of the specific contribution of sample $(x^T, x^I)$ within contrastive loss. Assume $(x^T, x^I)$ is assigned as the $n$-th pair in the $m$-th batch, in which the text and image data are embedded into matrix $U_m$ and $V_m$. Then $(x^T, x^I)$ serves as positive samples for each other in the $n$-th pairing loss $L_{T2I}(u_n, V_m; \theta)$ and $L_{I2T}(v_n, U_m; \theta)$ in (2). And $x^I$ and $x^T$ serve as negative samples in other pairing losses.

Through simple observation about (2), it can be noted that when the data serves as a positive sample, its influence can be explicitly isolated. However, its information is coupled with other data when acting as a negative sample, necessitating further analysis. We provide the derivation of the influence function for these two scenarios separately.

### 4.1  INFLUENCE AS POSITIVE SAMPLES

To quantify the impact of $x^T$ and $x^I$ as positive samples, ideally, we can retrain the model after removing the corresponding $n$-th pairing tasks, i.e., removing $L_{T2I}(u_n, V_m; \theta)$ and $L_{I2T}(v_n, U_m; \theta)$ in the loss function. Thus, following the idea of influence function, we can up-weight these two parts by $\epsilon$ and obtain an up-weighted loss function as the following with $\text{Pos}(x^T, x^I, \theta) = L_{T2I}(u_n, V_m; \theta) + L_{I2T}(v_n, U_m; \theta)$.

$$L_{\text{Total},\epsilon}(\theta) = \sum_{(U,V) \in \mathcal{B}} L_{\text{Batch}}(U, V; \theta) + \frac{\delta}{2} \|\theta\|_2^2 + \epsilon \cdot \text{Pos}((x^T, x^I); \theta).$$

And the parameters are obtained by $\hat{\theta}_\epsilon = \arg\min_\theta L_{\text{Total},\epsilon}(\theta)$. Then the influence function related to parameters can be deduced as:

$$\text{positive-IF}((x^T, x^I); \hat{\theta}) = -H_{\hat{\theta}}^{-1} \cdot \nabla_\theta \text{Pos}((x^T, x^I); \hat{\theta}). \tag{5}$$

where $H_{\hat{\theta}} = \nabla_\theta^2 \sum_{(U,V) \in \mathcal{B}} L_{\text{Batch}}(U, V; \hat{\theta}) + \delta I$ is the Hessian matrix at $\hat{\theta}$. The proof can be found in Section C.1.

**Extension to Multiple Samples.** The influence evaluation described above can be extended to a subset $\mathcal{D}^* \subset \mathcal{D}$. Let set $S$ to index the batches containing data from $\mathcal{D}^*$. For every $m \in S$, define an index set $E_m$ to specify the position of data from $\mathcal{D}^*$ within the $m$-th batch. We encapsulate the assigned results as $Seg = \{(m, E_m)|m \in S\}$. By employing a derivation method similar to that used for a single data point, we can obtain the parameter-related influence function for $\mathcal{D}^*$ by summing the influence as a position sample (5) for all samples in $\mathcal{D}^*$.

**Proposition 4.1.** *The influence function for dataset $\mathcal{D}^*$ serving as positive samples (positive-IF) can be approximated by*

$$positive\text{-}IF(\mathcal{D}^*, Seg; \hat{\theta}) = -H_{\hat{\theta}}^{-1} \cdot \nabla_\theta Pos(\mathcal{D}^*, Seg, \hat{\theta}),$$

*where*

$$Pos(\mathcal{D}^*, Seg; \hat{\theta}) = \sum_{m \in S} \sum_{n \in E_m} \left( L_{T2I}(u_n, V_m; \hat{\theta}) + L_{I2T}(v_n, U_m; \hat{\theta}) \right).$$

### 4.2 Influence as Negative Samples

In Section 4.1, we quantified the impact of $x^T$ and $x^I$ as positive samples by removing related pairing tasks. Next, we attempt to estimate their impact as negative samples by removing them from tasks where they serve as negative samples. To achieve this, we need to delve into the specific form of contrastive loss.

Take the text2image (T2I) loss for the $k$-th text embedding $u_k$ as the example, we first calculate its similarity with all image embeddings in the batch to form a similarity vector $S(u_k, V)$, which is then processed through a softmax layer $\sigma(\cdot)$ to yield a probability distribution. The $k$-th element indicates the probability of correctly pairing the text $u_k$ with its corresponding image: $[\sigma(S(u_k, V))]_k = \frac{e^{S_{k,k}}}{\sum_{j \in [B]} e^{S_{k,j}}}$, where $B$ is the batchsize. The model is encouraged to enhance the probability of correct pairing by minimizing the negative logarithm of this value. For $n \neq k$, $v_n$ serves as a negative sample in this task and appears in the $S_{k,n}$ term in the denominator. Thus, after removing the impact of $(x^T, x^I)$ as a negative sample from the $m$-th batch, the loss function corresponding to this batch should become:

$$L_{\text{T2I, -neg}}^m((x^T, x^I), S; \theta) = \sum_{\substack{k \in [B] \\ k \neq n}} -\log \frac{e^{S_{k,k}}}{\sum_{\substack{j \in [B] \\ j \neq n}} e^{S_{k,j}}} + Pos((x^T, x^I); \theta). \tag{6}$$

The original influence function method evaluates a data point's impact by adjusting its weight via a separate term in the loss function and getting the response function (4). In Contrastive Learning, however, the influence of data points as negative samples is coupled with information from other data, which can observed from (6). We will try to separate an influence term related to the data effect when it serves as a negative sample. Actually, the modification in (6) is analogous to eliminating the $n$-th row and column from the original similarity matrix. Leveraging the idea of deriving the influence function, we aim to develop a response function that converges to the target loss by up-weighting specific components.

Considering that similarities vectors are processed through the softmax layer, if we increase the similarity associated with $u_n$ and $v_n$ to a value approaching negative infinity, then after the exponential operation and the logarithmic function, the influence of $e^{S_{*,k}}$ and $e^{S_{n,*}}$ will become negligible. Mathematically, let $E_n$ be an $B \times B$ matrix such that its $n$-th column and the $n$-th row comprises ones, while all other entries are zero. We add the matrix $\log \zeta \times E_n$ to the similarity matrix. Then the loss function based on the revised similarity matrix becomes:

$$L_{\text{T2I},\zeta}^m((x^T, x^I), S; \theta) = \sum_{\substack{k \in [B] \\ k \neq n}} -\log \frac{e^{S_{k,k}}}{\sum_{j \in [B]} e^{S_{k,j}} + (\zeta - 1) \cdot e^{S_{k,n}}} + Pos((x^T, x^I); \theta). \tag{7}$$

We can easily see that as $\zeta$ approaches 0, the loss function $L_{\text{T2I},\zeta}^m$ in (7) converges to $L_{\text{T2I, -neg}}^m$ in (6). When $\zeta = 1$, the loss function equals the original one. To further separate this influence as

negative samples from the original loss function, we perform a Taylor expansion at $\zeta = 1$ and drop the $O((\zeta - 1)^2)$ term, then $L_{\text{T2I},\zeta}^m$ becomes

$$L_{\text{T2I}}^m(S;\theta) + (\zeta - 1) \cdot \sum_{\substack{k \in [B] \\ k \neq n}} \left( \frac{\sum_{j \in [B]} e^{S_{k,j}}}{e^{S_{k,n}}} \right) \xrightarrow{\zeta \to 0} L_{\text{T2I}}^m(S;\theta) - \sum_{\substack{k \in [B] \\ k \neq n}} \left( \frac{\sum_{j \in [B]} e^{S_{k,j}}}{e^{S_{k,n}}} \right),$$

and the left side is an estimation for (7). The minus term indicates the influence of $(x^T, x^I)$ as negative samples. By employing a similar method, one can obtain $L_{\text{I2T},\lambda}^m$ for the image2text part. Denote Neg $\left( (x^T, x^I); \theta \right)$ as

$$\text{Neg}\left( (x^T, x^I); \theta \right) = \sum_{\substack{k \in [B] \\ k \neq n}} \left( \frac{\sum_{j \in [B]} e^{S_{k,j}}}{e^{S_{k,n}}} + \frac{\sum_{j \in [B]} e^{S_{j,k}}}{e^{S_{n,k}}} \right),$$

Down-weighting the influence as a negative sample by $\zeta$ from 1 to 0, this influence in the loss function is then approximately eliminated. Then, the negative-influence function related to parameters can be deduced as:

$$\text{negative-IF}((x^T, x^I); \hat{\theta}) = -H_{\hat{\theta}}^{-1} \cdot \nabla_\theta \text{Neg}((x^T, x^I); \hat{\theta}).$$

Similar to the previous section, we can extend a single sample to a set of samples $\mathcal{D}^*$ and corresponding positional index Seg.

**Proposition 4.2.** *The influence function for dataset $\mathcal{D}^*$ serving as negative samples (negative-IF) can be approximated by*

$$\text{negative-IF}(\mathcal{D}^*, Seg; \hat{\theta}) = -H_{\hat{\theta}}^{-1} \cdot \nabla_\theta Neg(\mathcal{D}^*, Seg; \hat{\theta}),$$

*with*

$$Neg(\mathcal{D}^*, Seg; \hat{\theta}) = \sum_{m \in S} \sum_{k \in [B]/E_m} \left( \frac{\sum_{j \in [B]} e^{S_{k,j}}}{\sum_{n \in E_m} e^{S_{k,n}}} + \frac{\sum_{j \in [B]} e^{S_{j,k}}}{\sum_{n \in E_m} e^{S_{n,k}}} \right).$$

Combining Proposition 4.1 and 4.2 together, we then define our influence function method on contrastive learning(ECIF) as follows.

**Definition 4.3** (ECIF). The extended influence function for contrastive loss (ECIF) of the target dataset $\mathcal{D}^*$ with its position index set $Seg = \{(m, E_m) | m \in S\}$ is defined as

$$\text{ECIF}(\mathcal{D}^*, Seg; \hat{\theta}) \triangleq \left( \text{positive-IF}(\mathcal{D}^*, Seg; \hat{\theta}), \text{negative-IF}(\mathcal{D}^*, Seg; \hat{\theta}) \right).$$

With the assumption that the influence of data as positive and negative samples on model training can be linearly superimposed, we can employ ECIF to estimate the changes in model parameters resulting from data removal. We also give an upper bound on the error between the estimated influence given by ECIF and the actual influence obtained by model retraining in Appendix D for convex loss. We show that under certain scenarios, the approximation error becomes tolerable theoretically.

## 5 APPLICATIONS OF ECIF

We have proposed ECIF to evaluate the contribution of training data in contrastive learning. The ECIF method enables us to estimate the change in the learned parameters $\hat{\theta}$ if a training example pair is removed. Based on this, in this section, we will apply ECIF to two applications: misalignment detection and misprediction trace back.

### 5.1 MISALIGNMENT DETECTION

MLLMs typically assume a consistent alignment between all image-text pairs, and thus, misaligned data can lead to incorrect interpretations of these relationships, ultimately degrading model performance. Intuitively, given a high-quality validation data $D'$, if $D^*$ is a misaligned set, then the loss of $D'$ over the original model $\hat{\theta}$ should be greater than it over the model after deleting these misaligned data. And such a difference can be approximated by ECIF.

**Property 5.1.** *Considering a specific set $\mathcal{D}'$ with text and image embeddings $U'$ and $V'$, and a dataset $D^*$ to be removed, then we have*

$$L_{Batch}(U', V'; \hat{\theta}(-D^*)) - L_{Batch}(U', V'; \hat{\theta}) \approx \nabla L_{Batch}(U', V'; \hat{\theta})^{\mathrm{T}}(\hat{\theta}(-D^*) - \hat{\theta})$$

$$= -\nabla L_{Batch}(U', V'; \hat{\theta})^{\mathrm{T}} \cdot \left( \textit{positive-IF}(\mathcal{D}^*, Seg; \hat{\theta}) + \textit{negative-IF}(\mathcal{D}^*, Seg; \hat{\theta}) \right). \quad (8)$$

*where $\hat{\theta}(-D^*)$ is the optimal model for the loss eliminating $D^*$, positive-IF$(\mathcal{D}^*; Seg; \hat{\theta})$ and negative-IF$(\mathcal{D}^*, Seg; \hat{\theta})$ are obtained from Proposition 4.3 for $D^*$. We define term (8) as the task-related influence score, denoted as IS$(\mathcal{D}', \mathcal{D}^*, Seg; \hat{\theta})$.*

*Remark* 5.2. Task-related influence score estimates the actual impact of a data subset on a specific task. The sign of this score indicates whether the evaluated set $\mathcal{D}^*$ has a positive or negative impact on the correct execution of the test task, while the absolute value of the score represents the magnitude of this impact. Therefore, the misalignment detection problem is sum up as $\arg\max_{\mathcal{D}^* \subset \mathcal{D}} \mathrm{IS}(\mathcal{D}', \mathcal{D}^*, \mathrm{Seg}; \hat{\theta})$. See Appendix Algorithm 2 for details.

## 5.2 Misprediction Trace Back

From a transparency perspective, if the model makes prediction errors on certain tasks, the model trainers should be able to trace back to the samples in the training set associated with these erroneous predictions.

If we utilize the previous method for backtracking and choose the correct-labeled data which the model mispredicts to serve as the dataset $\mathcal{D}'$, then there is a significant possibility that the identified data are misaligned samples unrelated to the prediction errors. This is because, in the definition of task-relative IS, the term on the right side of the multiplication sign represents the change in model parameters. Even if certain samples are not related to the task we are tracing back, they may still have a high task-relative IS due to their substantial impact on the model parameters. Thus, compared to the above application, we need to constrain the change of model parameters.

To address this, consider imposing a constraint $\delta$ on the permissible changes in model parameters when tracing back from mispredicted data, while accounting for the process of upweighting the influence of samples as positive by $\epsilon$ and as negative by $\zeta$. Then we transform the trace back problem to identify which training example $x$ we should re-weight to most significantly impact the loss on the test sample set $\mathcal{D}'$ when given a small permissible change in model parameters $\delta$.

$$\arg\max_{x \in \mathcal{D}} \max_{\epsilon, \zeta} \left| L_{\mathrm{Batch}}(U', V'; \hat{\theta} + \Delta\hat{\theta}_{\epsilon, \zeta}(x)) - L_{\mathrm{Batch}}(U', V'; \hat{\theta}) \right| \quad \text{s.t.} \left\| \Delta\hat{\theta}_{\epsilon, \zeta}(x) \right\|^2 \leq \delta^2 \quad (9)$$

$$\approx \arg\max_{x \in \mathcal{D}} \max_{\epsilon, \zeta} |\nabla L_{\mathrm{Batch}}(U', V'; \hat{\theta})^{\mathrm{T}} \Delta\hat{\theta}_{\epsilon, \zeta}(x)| \quad \text{s.t.} \left\| \Delta\hat{\theta}_{\epsilon, \zeta}(x) \right\|^2 \leq \delta^2, \quad (10)$$

where $\Delta\hat{\theta}_{\epsilon, \zeta} = \epsilon \cdot \textit{positive-IF}(x; \hat{\theta}) + (\zeta - 1) \cdot \textit{negative-IF}(x; \hat{\theta})$ is the model parameter change estimated by ECIF when the influence of sample $x = (x^T, x^I)$ is upweighted by $\epsilon$ and $\zeta$.

**Proposition 5.3.** *Define $I = [\textit{positive-IF}(x), \textit{negative-IF}(x)]$. If the $2 \times 2$ matrix $I^{\mathrm{T}} \cdot I$ is irreversible, then equation (10) is equivalent to*

$$\arg\max_{x \in \mathcal{D}} \|\textit{negative-IF}(x; \hat{\theta})\|_2^{-1} \left| \nabla L_{Batch}(U', V'; \hat{\theta})^{\mathrm{T}} \cdot \textit{negative-IF}(x; \hat{\theta}) \right|.$$

*Else, $I^{\mathrm{T}} \cdot I$ is reversible, then (10) is equivalent to*

$$\arg\max_{x \in \mathcal{D}} \|\nabla L_{Batch}(U', V'; \hat{\theta})\|_2^{-1} \left| \nabla L_{Batch}(U', V'; \hat{\theta})^{\mathrm{T}} \cdot I \cdot \left[ I^{\mathrm{T}} \cdot I \right]^{-1} \cdot I^{\mathrm{T}} \cdot \nabla L_{Batch}(U', V'; \hat{\theta}) \right|.$$

The proposition above reduces the original argmax trace back problem to a simpler argmax problem. Consequently, we define the simplified argmax objective as a novel influence metric **relative-IS**. This metric, by adding constraints on parameter perturbations, helps us more accurately identify task-relevant samples. See Appendix Algorithm 4 for details.

# 6 EXPERIMENT

In our experiments, we will apply our above methods to tasks, including identifying influential data (harmful data and valuable data) for fine-tuning through the task-related influence score, mispredictions trace-back, and detecting misaligned data.

## 6.1 EXPERIMENTAL SETTINGS

**Datasets.**   We employ three datasets for utility and efficiency evaluation and the misprediction trace-back: *FGVC-Aircraft dataset* (Maji et al., 2013), *Food101 dataset* (Bossard et al., 2014), *Flowers102 dataset* (Nilsback & Zisserman, 2008). For the identifying influential data experiments, we include *Describable Textures Dataset(DTD) dataset* (Sharan et al., 2014) except for the above ones. For misalignment detection tasks, we use *Cifar-10dataset* (Krizhevsky, 2009), and *Imagenette*, a smaller subset of 10 easily classified classes from *Imagenet* (Deng et al., 2009).

**Algorithm.**   The tasks described below are direct implementations of the algorithms for the applications in the previous section. Algorithm 1 functions as the foundational algorithm, offering methods to calculate ECIF and providing model editing based on ECIF. Algorithm 2 and 3 compute task-related IS in Property 5.1 to evaluate samples, indicating both the direction and intensity of their impact on the task. Meanwhile, Algorithm 4 is for relative-IS in Prop. 5.3, which aids in tracing back specific samples.

**Ground Truth, Baselines and Evaluation Metric.**   We employ retraining as the ground truth, in which we finetune the CLIP from scratch after sample removal. We employ ECIF as the baseline. *ECIF*: This method is a direct implementation of Algorithm 1, utilizing positive and negative IF to modify the model for sample removal. We utilize two main evaluation metrics to assess our models: accuracy and runtime (RT). Accuracy evaluates the model's performance by measuring the proportion of correctly classified instances out of the total instances. Runtime, measured in seconds, assesses the time required for each method to update the model.

**Implementation Details.**   Our experiments utilized an Nvidia V100-32G GPU and 10 CPU cores with $64$ GB memory. For all experiments, we employ the CLIP model 'ViT-B/16' and LoRA few-shot learning. For utility evaluation, when testing our method on a random sample-removing task, $10\%$ samples are randomly removed. For valuable (harmful) samples, we remove $10\%$ of the valuable (harmful) data identified by ECIF. Each removal is repeated for 3 times with different seeds. See Appendix F.1 for details about other tasks.

## 6.2 UTILITY AND EFFICIENCY EVALUATION

We evaluate the utility and efficiency of ECIF for data evaluation, whose results are in Table 1. The results underscore the superior performance of ECIF compared to classical retraining. Notably, ECIF retains computational efficiency without sacrificing accuracy. We can easily observe that with random data removal, ECIF achieves an accuracy nearly equivalent to retraining (84.8784.87 compared to 84.9384.93) while significantly reducing runtime from 14.5914.59 seconds to 7.287.28 seconds on the Food101 dataset. A similar trend was observed in the Flowers102 dataset, where ECIF reduces runtime from 16.5916.59 seconds for retraining to 7.297.29 seconds, along with a modest 0.370.37 point improvement in accuracy. These findings demonstrate the ability of ECIF to save approximately 4040-50

When valuable data identified by ECIF are removed, the accuracy of both the retrained model and ECIF's edited version closely align, and both are significantly lower than those observed with random removal. This suggests that ECIF is capable of not only accurately editing the model but also effectively identifying influential data. Similar results can also be observed in the context of harmful data removal. See Appendix F.4 for the results on different numbers of removal samples.

## 6.3 IDENTIFYING INFLUENTIAL DATA FOR FINE-TUNING VIA TASK-RELATED IS

**Task-related IS can identify the most valuable data.** To numerically assess the precision of data valuation algorithms, we employ the brittleness test (Ilyas et al., 2022), which evaluates the al-

Table 1: Performance comparison of retraining and ECIF on different datasets.

| Sample | Method | FGVCAircraft | | Food101 | | Flowers102 | |
|--------|--------|--------------|--------------|--------------|--------------|--------------|--------------|
| | | Accuracy(%) | RT (second) | Accuracy(%) | RT (second) | Accuracy(%) | RT (second) |
| Random | Retrain | 23.07±0.29 | 19.57 | 84.93±0.17 | 14.59 | 68.16±0.22 | 16.59 |
| | ECIF | **22.77±0.09** | **7.60** | **84.87±0.24** | **7.28** | **68.53±0.12** | **7.29** |
| Valuable | Retrain | 22.93±0.33 | 15.56 | 84.80±0.16 | 15.88 | 68.23±0.33 | 16.43 |
| | ECIF | **22.73±0.09** | **5.95** | **84.86±0.05** | **6.27** | **68.26±0.12** | **6.52** |
| Harmful | Retrain | 23.50±0.11 | 22.40 | 84.83±0.05 | 14.59 | 68.00±0.16 | 16.09 |
| | ECIF | **23.02±0.07** | **6.26** | **84.90±0.01** | **6.22** | **68.30±0.01** | **6.27** |

gorithm's ability to accurately identify the most valuable data for a specific task. Our evaluation process is as follows: utilizing the validation set within Algorithm 2, we compute the task-related IS for each individual training data point. We then remove the top-$k$ valuable data points, with $k$ ranging from 5% to 30%, retrain the model multiple times using different random seeds, and assess the resultant change in overall model accuracy.

Results in Figure 1b reveal that removing valuable data identified by ECIF leads to a consistent decline in model accuracy, from 84.7 to 84.1. Conversely, random data removal triggers an increase in model accuracy once the removal proportion reaches 0.3. This suggests Food101 contains substantial noise, and our algorithm can effectively identify data points that genuinely enhance the model's predictive accuracy.

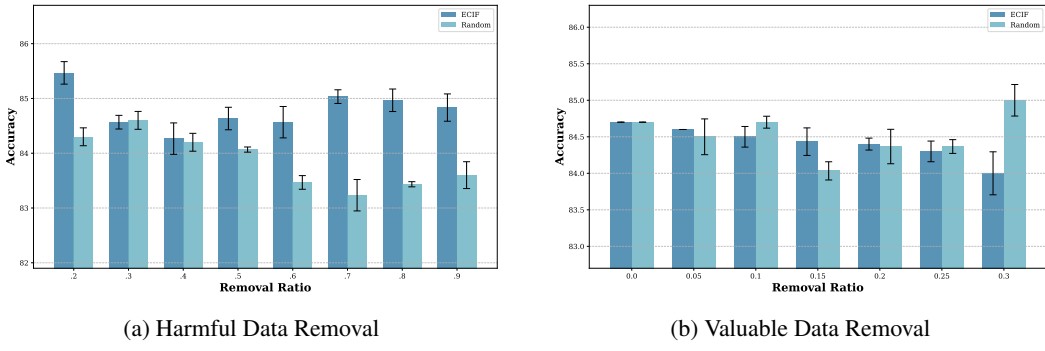

(a) Harmful Data Removal        (b) Valuable Data Removal

Figure 1: Accuracy after removing influential data by task-related IS on Food101 Dataset.

**Task-related IS can identify harmful data.** The influence analysis from Algorithm 2 identifies data pairs with negative task-related IS as *harmful* data for the task. To demonstrate the effectiveness of our algorithm in identifying detrimental data to specific tasks, we conducted experiments on several noisy datasets, such as Food101, and used the validation dataset in Algorithm 2.

We collected the harmful data identified by ECIF and then retrained the model multiple times with varying harmful data removal ratios and different random seeds. We compared its accuracy to that of a model retrained after randomly removing an equivalent number of data points. Results in Figure 1a demonstrate the effectiveness of our approach in improving model performance by eliminating harmful data using task-related IS. Figure 1a indicates that with varying proportions of harmful data removal, the accuracy of the retrained model consistently fluctuates around its original level. When 10% of harmful data is removed, accuracy increases by approximately 1%. Conversely, with random deletions, accuracy continues to decrease. This suggests that the accuracy improvement from removing harmful data with ECIF is not merely due to the removal action itself but rather because the removed data genuinely had a detrimental effect on model training. Additional results on other datasets are demonstrated in Appendix F.5.

## 6.4 VISUALIZATION OF MISPREDICTION TRACE BACK

We apply Algorithm 4 to identify training data that are most relevant to specific mispredicted test samples. In this process, we select samples in the test data on which the model made a misclassification. Using the relative IS, we can identify the training data with the highest influence on the

misprediction and visualize it. Table 2 shows the results of this misprediction trace-back process (see Appendix F.6 for additional results). Each pair of images compares a test sample with its most influential training counterpart. On the left, we show examples from the test set where the model produced incorrect predictions. On the right, the corresponding training data are shown, i.e., these data points hold the highest relative ISs in relation to the mispredicted test samples. This comparison helps shed light on how specific training samples may have contributed to the model's incorrect outputs. According to the visualization results, it can be observed that the samples traced back to the original task exhibit similarities in shape or texture with the original task.

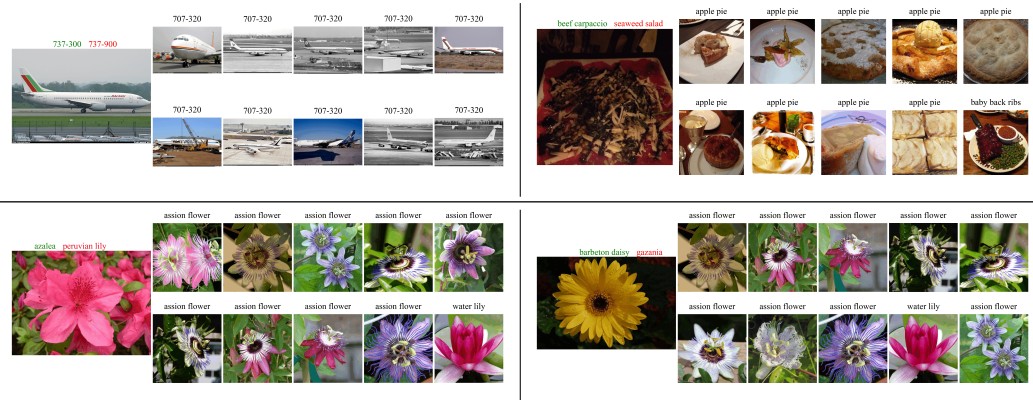

Table 2: Top-10 related training data traced by mispredicted data.

### 6.5 DATASET CLEANING: MISALIGNMENT DATA DETECTION

We employed the relative IF to detect misaligned data pairs. Regarding the selection of the validation dataset, we experimented with two approaches: randomly selecting samples from the gold dataset (Algorithm 2) and calculating based on the influence of the evaluated sample points (Algorithm 3), in which the test loss is defined as the CLIP score (Hessel et al., 2022) of the evaluated data pair.

We first mislabeled 10%-30% training samples and then identified the misaligned pairs by selecting those with the highest negative IS. These pairs are visualized in Table 3 (see Appendix F.7 for additional results). The visualization results reveal that the 8 data points with the highest IS are entirely within the mislabeled data in our training set. This suggests that our algorithm has effectively identified the noise data artificially introduced into the dataset.

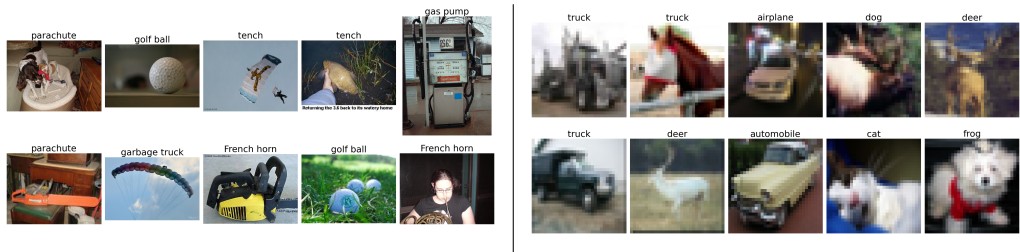

Table 3: Top-10 misaligned sample pairs in the 20% mislabeled training data.

## 7 CONCLUSION

In this paper, we introduced the Extended Influence Function for Contrastive Loss (ECIF), a novel method to quantify data valuation in MLLMs. ECIF provides a dual-perspective analysis of data points by considering both positive and negative samples, offering a more comprehensive understanding of their impact on model performance. By utilizing a closed-form approximation, ECIF eliminates the need for re-training, making it highly practical for large models. Our approach is applicable to enhancing fine-tuning, tracing mispredicted data, and detecting misaligned data, with results demonstrating its effectiveness in real-world tasks.

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

## A  ALGORITHM

---

**Algorithm 1** ECIF

---

1: **Input:** Training Dataset $\mathcal{D} = \{(x^T, x^I)\}$, dataset $\mathcal{D}^*$ to be evaluated, the parameters $\hat{\theta}$ which is involved in IF calculation in the model and the regularization parameter $\delta$.

2: Define $S(\cdot, \cdot)$ as the similarity score.

3: Compute the text embedding and image embedding for $\mathcal{D}^*$ as $u$ and $v$.

4: Random divide the training dataset $\mathcal{D}$ into MM batches and obtain the position index of $\mathcal{D}^*$ as $\text{Seg} \triangleq \{(m, E_m)|m \in S\}$.

5: Compute the influence term as positive and negative samples for the $m$-th batch in $S$ by:

$$\text{Pos}_m = \sum_{n \in E_m} \left( -\log \frac{e^{S(u_n, v_n)}}{\sum_{j=1}^N e^{S(u_n, v_j)}} - \log \frac{e^{S(v_n, u_n)}}{\sum_{j=1}^N e^{S(v_n, u_j)}} \right)$$

$$\text{Neg}_m = \sum_{i \in [N]/E_m} \left( \frac{\sum_{j=1}^N e^{S(u_i, v_j)}}{\sum_{n \in E_m} e^{S(u_i, v_n)}} + \frac{\sum_{j=1}^N e^{S(u_i, v_j)}}{\sum_{n \in E_m} e^{S(v_i, u_n)}} \right)$$

6: Compute the sum of the gradient of $\text{Pos}_m$ and $\text{Neg}_m$ as

$$\tilde{\text{Pos}} = \sum_{m \in S} \nabla_\theta \text{Pos}_m, \text{ and } \tilde{\text{Neg}} = \sum_{m \in S} \nabla_\theta \text{Neg}_m.$$

7: Compute the batch embedding for $\mathcal{D}$ as $\{B_m, m \in [M]\}$.

8: Compute the inverse Hessian matrix of the loss function with respect to $\hat{\theta}$ as

$$G = \left[ \sum_{m \in [M]} \nabla_\theta^2 L_{\text{Batch}}(B_m; \hat{\theta}) + \delta \cdot I \right]^{-1}$$

9: Compute the positive-IF$(\mathcal{D}^*, \text{Seg})$ and negative-IF$(\mathcal{D}^*, \text{Seg})$ as:

$$\text{positive-IF}(\mathcal{D}^*, \text{Seg}) = -G \cdot \tilde{\text{Pos}}, \quad \text{negative-IF}(\mathcal{D}^*, \text{Seg}) = -G \cdot \tilde{\text{Neg}}$$

10: Obtain the ECIF as

$$\text{ECIF}(\mathcal{D}^*, \mathcal{D}) = (\text{positive-IF}, \text{negative-IF}) \tag{11}$$

11: Edit model parameter to unlearn dataset $\mathcal{D}^*$ by

$$\tilde{\theta} = \hat{\theta} - \text{positive-IF} - \text{negative-IF}$$

12: **Return:** ECIF$(\mathcal{D}^*, \mathcal{D})$, Edited parameter $\tilde{\theta}$.

---

## B  ACCELERATION FOR INFLUENCE FUNCTION

**LOGRA.** For one layer, given the input $x_i$, output $x_o$ and the weight $W$, the forward and backward computation can be written as $x_o = W x_i$, $\text{vec}(\mathcal{D}W) = \sum_{t=1}^T x_{i,t} \otimes \mathcal{D}x_{o,t}$, $\mathcal{D}x_i = W^{\text{T}}\mathcal{D}x_o$, where $T$ denotes for the sequence dimension in language modeling, $\mathcal{D}$ the derivative with respect to the loss, $\otimes$ the Kronecker product, and $\text{vec}(\cdot)$ the vectorization operation. Observing gradient $\text{vec}(\mathcal{D}W)$ obtained during backpropagation is structured as a sum of Kronecker products between forward and backward activations, LOGRA imposes an additional Kronecker-product structure on the projection

---

**Algorithm 2** Task-related Influence Score Based on ECIF

---

1: **Input:** Training Dataset $\mathcal{D} = \{(x^T, x^I)\}$, dataset $\mathcal{D}^*$ to be evaluated, test dataset $\mathcal{D}'$, the parameters $\hat{\theta}$ which is involved in IF calculation in the model.
2: Compute the $\mathrm{ECIF}(\mathcal{D}^*, \mathcal{D}) = \big(\text{positive-IF}, \text{negative-IF}\big)$ by algorithm 1.
3: Compute the gradient of the batch loss function of the test data as

$$C = \sum_{(U,V)\in\mathcal{D}'} \nabla_\theta L_{\mathrm{Batch}}(U, V; \hat{\theta})$$

4: Compute the task-related influence score as

$$IS = C^{\mathrm{T}} \cdot \text{positive-IF} + C^{\mathrm{T}} \cdot \text{negative-IF}$$

5: **Return:** Task-related Influence Score $IS$.

---

**Algorithm 3** Self Influence Score Based on ECIF

---

1: **Input:** Training Dataset $\mathcal{D} = \{(x^T, x^I)\}$, dataset $\mathcal{D}^*$ to be evaluated, test dataset $\mathcal{D}'$, the parameters $\hat{\theta}$ which is involved in IF calculation in the model.
2: Compute the $\mathrm{ECIF}(\mathcal{D}^*, \mathcal{D}) = \big(\text{positive-IF}, \text{negative-IF}\big)$ by algorithm 1.
3: Compute the gradient of the batch loss function of the test data as

$$C = \frac{1}{|\mathcal{D}'|} \sum_{(x^T,x^I)\in\mathcal{D}'} \nabla_\theta - \log \frac{u^{\mathrm{T}} \cdot v}{\|u\| \cdot \|v\|},$$

where $u$ and $v$ is the embedding for $x^T$ and $x^I$
4: Compute the task-related influence score as

$$IS = C^{\mathrm{T}} \cdot \text{positive-IF} + C^{\mathrm{T}} \cdot \text{negative-IF}$$

5: **Return:** Task-related Influence Score $IS$.

---

**Algorithm 4** Relative Influence Score Based on ECIF

---

1: **Input:** Training Dataset $\mathcal{D} = \{(x^T, x^I)\}$, dataset $\mathcal{D}^*$ to be evaluated, test dataset $\mathcal{D}'$.
2: Compute the $\mathrm{ECIF}(\mathcal{D}^*, \mathcal{D}) = (\text{positive-IF}, \text{negative-IF})$ by algorithm 1.
3: Compute the gradient of the batch loss function of the test data as

$$C = \sum_{(U,V)\in\mathcal{D}'} \nabla_\theta L_{\mathrm{Batch}}(U, V; \hat{\theta})$$

4: **if** positive-IF is parallel to negative-IF **then**
5:     Compute the relative-IS as

$$\text{relative-IS} = \|\text{positive-IF}\|^{-1} \left| C^{\mathrm{T}}\text{negative-IF} \right|$$

6: **else** {positive-IF is not parallel to negative-IF}
7:     Define $I = [\text{positive-IF}, \text{negative-IF}]$.
8:     Compute the relative-IS as

$$\text{relative-IS} = \|C\|^{-1} \left| C^{\mathrm{T}} I \left[ I^{\mathrm{T}} \cdot I \right]^{-1} I^{\mathrm{T}} C \right|$$

9: **end if**
10: **Return:** relative-IS.

---

---

**Algorithm 5** Relative Influence Score Based on ECIF

---

1: **Input:** Training Dataset $\mathcal{D} = \{(x^T, x^I)\}$, dataset $\mathcal{D}^*$ to be evaluated, test dataset $\mathcal{D}'$.
2: Compute the $\text{ECIF}(\mathcal{D}^*, \mathcal{D}) = (\text{positive-IF}, \text{negative-IF})$ by algorithm 1.
3: Compute the gradient of the batch loss function of the test data as

$$C = \frac{1}{|\mathcal{D}'|} \sum_{(x^T, x^I) \in \mathcal{D}'} \nabla_\theta - \log \frac{u^\mathrm{T} \cdot v}{\|u\| \cdot \|v\|},$$

where $u$ and $v$ is the embedding for $x^T$ and $x^I$
4: **if** positive-IF is parallel to negative-IF **then**
5:    Compute the relative-IS as

$$\text{relative-IS} = \|\text{positive-IF}\|^{-1} \left| C^\mathrm{T} \text{negative-IF} \right|$$

6: **else** {positive-IF is not parallel to negative-IF}
7:    Define $I = [\text{positive-IF}, \text{negative-IF}]$.
8:    Compute the relative-IS as

$$\text{relative-IS} = \|C\|^{-1} \left| C^\mathrm{T} I \left[ I^\mathrm{T} \cdot I \right]^{-1} I^\mathrm{T} C \right|$$

9: **end if**
10: **Return:** relative-IS.

---

matrix P as follows:

$$P\text{vec}(\mathcal{D}W) \triangleq (P_i \otimes P_o)\text{vec}(\mathcal{D}W) = \sum_{t=1}^{T}(P_i \otimes P_o)(x_{i,t} \otimes Dx_{o,t}) = \sum_{t=1}^{T} P_i x_{i,t} \otimes P_o Dx_{o,t}$$

where $P_i$ is the projection matrix on the input and $P_o$ is that on the backward activations, and $P \triangleq P_i \otimes P_o$.

# C   INFLUENCE FUNCTION IN CONTRASTIVE LEARNING

## C.1   INFLUENCE FUNCTION FOR POSITIVE SAMPLES.

We first consider the influence function for positive samples.

**Single Data Pair Version.**    To quantify the impact of $x^T$ and $x^I$ as positive samples, we first define $L_{T2I}(u_n, V_m; \theta) + L_{I2T}(v_n, U_m; \theta)$ as $\text{Pos}((x^T, x^I); \theta)$. Following the idea of influence function, we can up-weight these two parts by $\epsilon$ and obtain an up-weighted loss function as

$$L_{\text{Total},\epsilon}(\theta) = \sum_{(U,V) \in \mathcal{B}} L_{\text{Batch}}(U, V; \theta) + \frac{\delta}{2}\|\theta\|_2^2 + \epsilon \cdot \text{Pos}((x^T, x^I); \theta).$$

And the parameters are obtained by $\hat{\theta}_\epsilon = \arg\min_\theta L_{\text{Total},\epsilon}(\theta)$. From this minimizing condition, we have

$$\sum_{(U,V) \in \mathcal{B}} \nabla_\theta L_{\text{Batch}}(U, V; \hat{\theta}_\epsilon) + \epsilon \cdot \nabla_\theta \text{Pos}((x^T, x^I); \hat{\theta}_\epsilon) = 0$$

Perform a Taylor expand at $\theta = \hat{\theta}$, we have

$$\sum_{(U,V) \in \mathcal{B}} \nabla_\theta L_{\text{Batch}}(U, V; \hat{\theta}) + \epsilon \cdot \nabla_\theta \text{Pos}((x^T, x^I); \hat{\theta}) + \sum_{(U,V) \in \mathcal{B}} \nabla_\theta^2 L_{\text{Batch}}(U, V; \hat{\theta}) \cdot \left(\hat{\theta}_\epsilon - \hat{\theta}\right) \approx 0$$

Because $\hat{\theta}$ minimizes $\sum_{(U,V) \in \mathcal{B}} L_{\text{Batch}}(U, V; \hat{\theta})$, the first term in the above equation equals 0.

$$\text{positive-IF}((x^T, x^I); \hat{\theta}) \triangleq \left.\frac{\mathrm{d}\hat{\theta}_\epsilon}{\mathrm{d}\epsilon}\right|_{\epsilon=0} = -H_{\hat{\theta}}^{-1} \cdot \nabla_\theta \text{Pos}((x^T, x^I); \hat{\theta})$$

where $H_{\hat{\theta}} = \nabla_\theta^2 \sum_{(U,V) \in \mathcal{B}} L_{\text{Batch}}(U, V; \hat{\theta}) + \delta I$ is the Hessian matrix at $\hat{\theta}$.

**Extension to Multiple Samples.** The influence evaluation described above can be extended to a subset $\mathcal{D}^* \subset \mathcal{D}$. Let set $S$ to index the batches containing data from $\mathcal{D}^*$. For every $m \in S$, define an index set $E_m$ to specify the position of data from $\mathcal{D}^*$ within the $m$-th batch. We encapsulate the assigned results as Seg $= \{(m, E_m)|m \in S\}$. By employing a derivation method similar to that used for a single data point, we can obtain the parameter-related influence function for $\mathcal{D}^*$ by summing the influence as a position sample (5) for all samples in $\mathcal{D}^*$.

**Proposition C.1.** *The influence function for dataset $\mathcal{D}^*$ serving as positive sample (positive-IF) is*

$$positive\text{-}IF(\mathcal{D}^*, Seg; \hat{\theta}) = -H_{\hat{\theta}}^{-1} \cdot \nabla_\theta Pos(\mathcal{D}^*, Seg; \hat{\theta})$$

*where*

$$Pos(\mathcal{D}^*, Seg; \hat{\theta}) = \sum_{m \in S} \sum_{n \in E_m} \left( L_{T2I}(u_n, V_m; \hat{\theta}) + L_{I2T}(v_n, U_m; \hat{\theta}) \right)$$

*Proof.* Seg $= \{(m, E_m)|m \in S\}$, for $m \in S$, $U_m$, $V_m$ are the text and image embedding for the $m$-th batch, respectively. For $n \in E_m$, $u_n$ and $v_n$ are embeddings for a single data pair in $m$-th batch, which is included in the dataset to be evaluated $\mathcal{D}^*$. Define $Pos(\mathcal{D}^*, Seg; \theta)$ as

$$\text{Pos}(\mathcal{D}^*, \text{Seg}; \theta) = \sum_{m \in S} \sum_{n \in E_m} (L_{T2I}(u_n, V_m; \theta) + L_{I2T}(v_n, U_m; \theta))$$

Following the idea of influence function, we can up-weight these by $\epsilon$ and obtain an up-weighted loss function as

$$L_{\text{Total},\epsilon}(\theta) = \sum_{(U,V) \in \mathcal{B}} L_{\text{Batch}}(U, V; \theta) + \frac{\delta}{2}\|\theta\|_2^2 + \epsilon \cdot \text{Pos}(\mathcal{D}^*, \text{Seg}; \theta).$$

And the parameters are obtained by $\hat{\theta}_\epsilon = \arg\min_\theta L_{\text{Total},\epsilon}(\theta)$. From this minimizing condition, we have

$$\sum_{(U,V) \in \mathcal{B}} \nabla_\theta L_{\text{Batch}}(U, V; \hat{\theta}_\epsilon) + \epsilon \cdot \nabla_\theta \text{Pos}(\mathcal{D}^*, \text{Seg}; \hat{\theta}_\epsilon) = 0$$

Perform a Taylor expand at $\theta = \hat{\theta}$, we have

$$\sum_{(U,V) \in \mathcal{B}} \nabla_\theta L_{\text{Batch}}(U, V; \hat{\theta}) + \epsilon \cdot \nabla_\theta \text{Pos}(\mathcal{D}^*, \text{Seg}; \hat{\theta}) + \sum_{(U,V) \in \mathcal{B}} \nabla_\theta^2 L_{\text{Batch}}(U, V; \hat{\theta}) \cdot \left( \hat{\theta}_\epsilon - \hat{\theta} \right) \approx 0$$

Because $\hat{\theta}$ minimizes $\sum_{(U,V) \in \mathcal{B}} L_{\text{Batch}}(U, V; \hat{\theta})$, the first term in the above equation equals 0.

$$\text{positive-IF}(\mathcal{D}^*, \text{Seg}; \hat{\theta}) \triangleq \frac{\mathrm{d}\hat{\theta}_\epsilon}{\mathrm{d}\epsilon}\bigg|_{\epsilon=0} = -H_{\hat{\theta}}^{-1} \cdot \nabla_\theta \text{Pos}(\mathcal{D}^*, \text{Seg}; \hat{\theta})$$

where $H_{\hat{\theta}} = \nabla_\theta^2 \sum_{(U,V) \in \mathcal{B}} L_{\text{Batch}}(U, V; \hat{\theta}) + \delta I$ is the Hessian matrix at $\hat{\theta}$. $\qquad\square$

## C.2 INFLUENCE FUNCTION FOR NEGATIVE SAMPLES.

Then, we come to derive the influence function for the negative sample.

In this part, we will illustrate how we give an approximation function for the loss function in which the influence as a negative sample of the data we are considering is removed. With the help of Taylor expansion, this influence is separated into a single term in this approximation function, and we can achieve this by removing this term from the original loss function.

After removing the impact of $(x^T, x^I)$ as a negative sample from the $m$-th batch, the loss function corresponding to this batch should become:

$$L_{T2I,\text{-neg}}^m((x^T, x^I), S; \theta) = \sum_{\substack{k \in [B] \\ k \neq n}} \frac{e^{S_{k,k}}}{\sum_{\substack{j \in [B] \\ j \neq n}} e^{S_{k,j}}} + \text{Pos}((x^T, x^I); \theta). \tag{12}$$

Mathematically, let $E_n$ be an $B \times B$ matrix such that its $n$-th column and the $n$-th row comprises ones, while all other entries are zero. We add the matrix $\log \zeta \times E_n$ to the similarity matrix. Then $S_{*,n}$ becomes $S_{*,n} + \log \zeta$. The loss function based on the revised similarity matrix becomes:

$$L_{\text{T2I},\lambda}^m((x^T, x^I), S; \theta) = \sum_{\substack{k \in [B] \\ k \neq n}} - \log \frac{e^{S_{k,k}}}{\sum_{\substack{j \in [B] \\ j \neq n}} e^{S_{k,j}} + e^{\log \zeta} \cdot e^{S_{k,n}}} + \text{Pos}\left((x^T, x^I); \theta\right)$$

$$= \sum_{\substack{k \in [B] \\ k \neq n}} - \log \frac{e^{S_{k,k}}}{\sum_{\substack{j \in [B] \\ j \neq n}} e^{S_{k,j}} + \zeta \cdot e^{S_{k,n}}} + \text{Pos}\left((x^T, x^I); \theta\right)$$

$$= \sum_{\substack{k \in [B] \\ k \neq n}} - \log \frac{e^{S_{k,k}}}{\sum_{j \in [B]} e^{S_{k,j}} + (\zeta - 1) \cdot e^{S_{k,n}}} + \text{Pos}\left((x^T, x^I); \theta\right)$$

We can easily see that as $\zeta$ approaches $0$, the loss function $L_{\text{T2I},\zeta}^m$ converges to $L_{\text{T2I, -neg}}^m$ in (12). When $\zeta = 1$, the loss function equals the original one. To separate the change in the $\zeta$ approaching $0$ from $1$ process, we perform a Taylor expansion at $\zeta = 0$ and drop the $O((\zeta - 1)^2)$ term, then $L_{\text{T2I},\zeta}^m$ becomes

$$\sum_{\substack{k \in [B] \\ k \neq n}} - \log \frac{e^{S_{k,k}}}{\sum_{j \in [B]} e^{S_{k,j}}} + (\zeta - 1) \cdot \sum_{\substack{k \in [B] \\ k \neq n}} \left( \frac{\sum_{j \in [B]} e^{S_{k,j}}}{e^{S_{k,n}}} \right) + O((\zeta - 1)^2) + \text{Pos}((x^T, x^I); \theta).$$

And by setting $\zeta = 0$, the loss function $L_{\text{T2I},0}^m$ indicates that the influence of $(x^T, x^I)$ when it serves as the negative sample is fully removed from the training process.

$$L_{\text{T2I},0}^m = \sum_{\substack{k \in [B] \\ k \neq n}} - \log \frac{e^{S_{k,k}}}{\sum_{j \in [B]} e^{S_{k,j}}} + (0 - 1) \cdot \sum_{\substack{k \in [B] \\ k \neq n}} \left( \frac{\sum_{j \in [B]} e^{S_{k,j}}}{e^{S_{k,n}}} \right) + \text{Pos}((x^T, x^I); \theta)$$

$$= L_{\text{T2I}}^m(S; \theta) - \sum_{\substack{k \in [B] \\ k \neq n}} \left( \frac{\sum_{j \in [B]} e^{S_{k,j}}}{e^{S_{k,n}}} \right).$$

**Single Data Pair Version.** From above discussion, to quantify the impact of $x^T$ and $x^I$ as negative samples, we first define $\text{Neg}\left((x^T, x^I); \theta\right)$ as

$$\text{Neg}\left((x^T, x^I); \theta\right) = \sum_{\substack{k \in [B] \\ k \neq n}} \left( \frac{\sum_{j \in [B]} e^{S_{k,j}}}{e^{S_{k,n}}} + \frac{\sum_{j \in [B]} e^{S_{j,k}}}{e^{S_{n,k}}} \right),$$

Down-weighting the influence as a negative sample by $\zeta$ from $1$ to $0$, this influence in the loss function is then approximately eliminated. In this process, the loss function becomes

$$L_{\text{Total},\zeta}(\theta) = \sum_{(U,V) \in \mathcal{B}} L_{\text{Batch}}(U, V; \theta) + \frac{\delta}{2} \|\theta\|_2^2 + (\zeta - 1) \cdot \text{Neg}((x^T, x^I); \theta).$$

And the parameters are obtained by $\hat{\theta}_\zeta = \arg\min_\theta L_{\text{Total},\zeta}(\theta)$. From this minimizing condition, we have

$$\sum_{(U,V) \in \mathcal{B}} \nabla_\theta L_{\text{Batch}}(U, V; \hat{\theta}_\zeta) + (\zeta - 1) \cdot \nabla_\theta \text{Neg}((x^T, x^I); \hat{\theta}_\zeta) = 0$$

Perform a Taylor expand at $\theta = \hat{\theta}$, we have

$$\sum_{(U,V) \in \mathcal{B}} \nabla_\theta L_{\text{Batch}}(U, V; \hat{\theta}) + (\zeta - 1) \cdot \nabla_\theta \text{Neg}((x^T, x^I); \hat{\theta}) + \sum_{(U,V) \in \mathcal{B}} \nabla_\theta^2 L_{\text{Batch}}(U, V; \hat{\theta}) \cdot \left( \hat{\theta}_\zeta - \hat{\theta} \right) \approx 0$$

Because $\hat{\theta}$ minimizes $\sum_{(U,V)\in\mathcal{B}} L_{\text{Batch}}(U,V;\hat{\theta})$, the first term in the above equation equals 0. Then

$$\hat{\theta}_\zeta - \hat{\theta} = -(\zeta - 1) \cdot H_{\hat{\theta}}^{-1} \cdot \nabla_\theta \text{Neg}((x^T, x^I); \hat{\theta})$$

where $H_{\hat{\theta}} = \nabla_\theta^2 \sum_{(U,V)\in\mathcal{B}} L_{\text{Batch}}(U,V;\hat{\theta}) + \delta I$ is the Hessian matrix at $\hat{\theta}$.

$$\text{negative-IF}((x^T, x^I); \hat{\theta}) \triangleq \left. \frac{d\hat{\theta}_\zeta}{d\zeta} \right|_{\zeta=0} = -H_{\hat{\theta}}^{-1} \cdot \nabla_\theta \text{Neg}((x^T, x^I); \hat{\theta})$$

**Extension to Multiple Samples.** Then, we extend the above influence evaluation to a subset $\mathcal{D}^* \subset \mathcal{D}$. Let set $S$ to index the batches containing data from $\mathcal{D}^*$. For every $m \in S$, define an index set $E_m$ to specify the position of data from $\mathcal{D}^*$ within the $m$-th batch. We encapsulate the assigned results as $\text{Seg} = \{(m, E_m)|m \in S\}$. By employing a derivation method similar to that used for a single data point, we can obtain the parameter-related influence function for $\mathcal{D}^*$.

**Proposition C.2.** *The influence function for dataset $\mathcal{D}^*$ serving as negative sample (negative-IF) is*

$$Neg(\mathcal{D}^*, Seg; \theta) = \sum_{m \in S} \sum_{k \in [B]/E_m} \left( \frac{\sum_{j \in [B]} e^{S_{k,j}}}{\sum_{n \in E_m} e^{S_{k,n}}} + \frac{\sum_{j \in [B]} e^{S_{j,k}}}{\sum_{n \in E_m} e^{S_{n,k}}} \right)$$

*And*

$$\text{negative-IF}(\mathcal{D}^*, Seg; \hat{\theta}) = -H_{\hat{\theta}}^{-1} \cdot \nabla_\theta Neg(\mathcal{D}^*, Seg; \hat{\theta}).$$

*Proof.* $\text{Seg} = \{(m, E_m)|m \in S\}$, for $m \in S$, $U_m, V_m$ are the text and image embedding for the $m$-th batch, respectively. For $n \in E_m$, $u_n$ and $v_n$ are embeddings for a single data pair in $m$-th batch, which is included in the dataset to be evaluated $\mathcal{D}^*$.

**Step 1**. Noting the data in $\mathcal{D}^*$ may come from different batches and multiple data from one batch, then we firstly derive the loss function approximation with separated negative sample influence removed.

For the $m$-th batch, $m \in S$, after removing the impact of the data indexed by $E_m$ as a negative sample, the loss function corresponding to this batch should become:

$$L_{\text{T2I, -neg}}^m((x^T, x^I), S; \theta) = \sum_{\substack{k \in [B] \\ k \notin E_m}} \frac{e^{S_{k,k}}}{\sum_{\substack{j \in [B] \\ j \notin E_m}} e^{S_{k,j}}} + \text{Pos}((x^T, x^I); \theta). \tag{13}$$

Then, for $n \in E_m$, let $E_n$ be an $B \times B$ matrix such that its $n$-th column and the $n$-th row comprises ones, while all other entries are zero. We add the matrix $\log \zeta \times E_n$ to the similarity matrix. Then, the loss function based on the revised similarity matrix becomes:

$$L_{\text{T2I},\lambda}^m((x^T, x^I), S; \theta) = \sum_{\substack{k \in [B] \\ k \notin E_m}} -\log \frac{e^{S_{k,k}}}{\sum_{j \in [B]} e^{S_{k,j}} + (\zeta - 1) \cdot \sum_{n \in E_m} e^{S_{k,n}}} + \text{Pos}((x^T, x^I); \theta).$$

We can easily see that as $\zeta$ approaches 0, the loss function $L_{\text{T2I},\zeta}^m$ converges to $L_{\text{T2I, -neg}}^m$ in (12). When $\zeta = 1$, the loss function equals the original one. To separate the change in the $\zeta$ approaching 0 from 1 process, we perform a Taylor expansion at $\zeta = 0$ and drop the $O((\zeta - 1)^2)$ term, then $L_{\text{T2I},\zeta}^m$ becomes

$$\sum_{\substack{k \in [B] \\ k \notin E_m}} -\log \frac{e^{S_{k,k}}}{\sum_{j \in [B]} e^{S_{k,j}}} + (\zeta - 1) \cdot \sum_{\substack{k \in [B] \\ k \notin E_m}} \left( \frac{\sum_{j \in [B]} e^{S_{k,j}}}{\sum_{n \in E_m} e^{S_{k,n}}} \right) + O((\zeta - 1)^2) + \text{Pos}((x^T, x^I); \theta).$$

And by setting $\zeta = 0$, the loss function $L_{\text{T2I},0}^m$ indicates that the influence of $(x^T, x^I)$ when it serves as the negative sample is fully removed from the training process.

$$L_{\text{T2I},0}^m = \sum_{\substack{k \in [B] \\ k \notin E_m}} - \log \frac{e^{S_{k,k}}}{\sum_{j \in [B]} e^{S_{k,j}}} + (0 - 1) \cdot \sum_{\substack{k \in [B] \\ k \notin E_m}} \left( \frac{\sum_{j \in [B]} e^{S_{k,j}}}{\sum_{n \in E_m} e^{S_{k,n}}} \right) + \text{Pos}((x^T, x^I); \theta)$$

$$= L_{\text{T2I}}^m(S; \theta) - \sum_{\substack{k \in [B] \\ k \notin E_m}} \left( \frac{\sum_{j \in [B]} e^{S_{k,j}}}{\sum_{n \in E_m} e^{S_{k,n}}} \right)$$

By down-weighting the influence of $\mathcal{D}^*$ as negative samples by $\zeta$, the total loss function becomes

$$L_{\text{Total},\zeta}(\theta) = \sum_{(U,V) \in \mathcal{B}} L_{\text{Batch}}(U, V; \theta) + \frac{\delta}{2} \|\theta\|_2^2 + (\zeta - 1) \cdot \sum_{m \in S} \sum_{k \in [B]/E_m} \left( \frac{\sum_{j \in [B]} e^{S_{k,j}}}{\sum_{n \in E_m} e^{S_{k,n}}} \right)$$

Then denote $\text{Neg}(\mathcal{D}^*, \text{Seg}; \theta)$ as

$$\text{Neg}(\mathcal{D}^*, \text{Seg}; \theta) = \sum_{m \in S} \sum_{k \in [B]/E_m} \left( \frac{\sum_{j \in [B]} e^{S_{k,j}}}{\sum_{n \in E_m} e^{S_{k,n}}} + \frac{\sum_{j \in [B]} e^{S_{j,k}}}{\sum_{n \in E_m} e^{S_{n,k}}} \right)$$

And the loss function with the negative-sample influence of $\mathcal{D}^*$ explicitly removed is

$$L_{\text{Total},0}(\theta) = \sum_{(U,V) \in \mathcal{B}} L_{\text{Batch}}(U, V; \theta) + \frac{\delta}{2} \|\theta\|_2^2 - \text{Neg}(\mathcal{D}^*, \text{Seg}; \theta)$$

**Step 2**. The parameters are obtained by $\hat{\theta}_\zeta = \arg \min_\theta L_{\text{Total},\zeta}(\theta)$. From this minimizing condition, we have

$$\sum_{(U,V) \in \mathcal{B}} \nabla_\theta L_{\text{Batch}}(U, V; \hat{\theta}_\zeta) + (\zeta - 1) \cdot \nabla_\theta \text{Neg}(\mathcal{D}^*, \text{Seg}; \hat{\theta}) = 0$$

Perform a Taylor expand at $\theta = \hat{\theta}$, we have

$$\sum_{(U,V) \in \mathcal{B}} \nabla_\theta L_{\text{Batch}}(U, V; \hat{\theta}) + (\zeta - 1) \cdot \nabla_\theta \text{Neg}(\mathcal{D}^*, \text{Seg}; \hat{\theta}) + \sum_{(U,V) \in \mathcal{B}} \nabla_\theta^2 L_{\text{Batch}}(U, V; \hat{\theta}) \cdot \left( \hat{\theta}_\zeta - \hat{\theta} \right) \approx 0$$

Because $\hat{\theta}$ minimizes $\sum_{(U,V) \in \mathcal{B}} L_{\text{Batch}}(U, V; \hat{\theta})$, the first term in the above equation equals 0.

$$\text{negative-IF}(\mathcal{D}^*, \text{Seg}; \hat{\theta}) \triangleq \left. \frac{d\hat{\theta}_\zeta}{d\zeta} \right|_{\zeta=0} = -H_{\hat{\theta}}^{-1} \cdot \nabla_\theta \text{Neg}(\mathcal{D}^*, \text{Seg}; \hat{\theta})$$

where $H_{\hat{\theta}} = \nabla_\theta^2 \sum_{(U,V) \in \mathcal{B}} L_{\text{Batch}}(U, V; \hat{\theta}) + \delta I$ is the Hessian matrix at $\hat{\theta}$. $\qquad \square$

# D  APPROXIMATION ERROR BOUND

In the previous discussion, we have established that when applying the influence function method to contrastive learning, it is impractical to design a sample-specific up-weighting scheme that approximates the corresponding loss function resulting from the removal of a single pair in the batch without affecting the remaining data. Therefore, based on the previous derivation, we provide an estimation function $L^-$ for this loss function. Consider the dataset $\mathcal{D}^*$, define

$$L'(\mathcal{D}^*, \text{Seg}; \theta) \triangleq \text{Pos}(\mathcal{D}^*, \text{Seg}; \theta) + \cdot \text{Neg}(\mathcal{D}^*, \text{Seg}; \theta),$$

Then the loss function with the influence of $\mathcal{D}^*$ removed becomes

$$L^-(\mathcal{B}, \mathcal{D}^*, \text{Seg}; \theta) = L_{Total}(\mathcal{B}; \theta) - L'(\mathcal{D}^*, \text{Seg}; \theta). \tag{14}$$

Equation (14) is based on the assumption that the influence of data acting as positive and negative samples on model parameters can be linearly superimposed, and we can leverage ECIF to edit the

model based on the following corollary. This approach enables us to achieve the unlearning or updating of specific data without the need to remove data and retrain the model.

Assume $\hat{\theta} = \arg\min L_{Total}$ is the original model parameter, and $\hat{\theta}(-\mathcal{D}^*)$ is the minimizer of $L^-$, which is obtained from retraining. Denote $\theta_{if}(-\mathcal{D}^*)$ as the updated model with the influence of $\mathcal{D}^*$ removed and is obtained by the ECIF method, which is an estimation for $\hat{\theta}(-\mathcal{D}^*)$. Because we concentrate on $\mathcal{D}^*$, we omit the Seg in the above definitions for short.

In this part, we will study the error between the estimated influence given by the ECIF method and retraining. We use the parameter changes as the evaluation metric:

$$\left| \left( \theta_{if}(-\mathcal{D}^*) - \hat{\theta} \right) - \left( \hat{\theta}(-\mathcal{D}^*) - \hat{\theta} \right) \right| = \left| \theta_{if}(-\mathcal{D}^*) - \hat{\theta}(-\mathcal{D}^*) \right| \tag{15}$$

Before our main theorem of the upper bound for equation (15), we need to prove corollaries and make some assumptions.

**Proposition D.1.** *Assume that influence as positive sample and as negative sample can be linearly superposed. Then when the influence of dataset $\mathcal{D}^*$ as positive sample is up-weighted by $\epsilon$ and that as negative sample is up-weighted by $\zeta$, then the loss function become*

$$L^-(\mathcal{D}^*, Seg; \theta; \epsilon, \zeta) \triangleq L_{Total}(\mathcal{B}; \theta) + \epsilon \cdot \nabla_\theta Pos(\mathcal{D}^*, Seg; \hat{\theta}) + (\zeta - 1) \cdot Neg(\mathcal{D}^*, Seg; \hat{\theta})$$

*And corresponding parameters $\theta_{\epsilon,\zeta}$ are defined as*

$$\hat{\theta}_{\epsilon,\zeta}(-\mathcal{D}^*) = \arg\min_\theta L^-(\mathcal{D}^*, Seg; \theta; \epsilon, \zeta)$$

*The approximation of $\hat{\theta}_{\epsilon,\zeta}(-\mathcal{D}^*)$ is derived as*

$$\hat{\theta}_{\epsilon,\zeta}(\mathcal{D}^*) \approx \theta_{\epsilon,\zeta}(\mathcal{D}^*) = \hat{\theta} - H_{\hat{\theta}}^{-1} \cdot \left( \frac{\sqrt{2}}{2} \cdot \nabla_\theta Pos(\mathcal{D}^*, Seg; \hat{\theta}) + \frac{\sqrt{2}}{2} \cdot \nabla_\theta Neg(\mathcal{D}^*, Seg; \hat{\theta}) \right) \tag{16}$$

**Property D.2.** *Assume that influence as positive sample and as negative sample can be linearly superposed. Then when the influence of dataset $\mathcal{D}^*$ as positive sample is up-weighted by $\epsilon$ and that as negative sample is up-weighted by $\zeta$, then the loss function become*

$$L^-(\mathcal{D}^*, Seg; \theta; \epsilon, \zeta) \triangleq L_{Total}(\mathcal{B}; \theta) + \epsilon \cdot \nabla_\theta Pos(\mathcal{D}^*, Seg; \hat{\theta}) + (\zeta - 1) \cdot Neg(\mathcal{D}^*, Seg; \hat{\theta})$$

*And corresponding parameters $\theta_{\epsilon,\zeta}$ are defined as*

$$\hat{\theta}_{\epsilon,\zeta}(-\mathcal{D}^*) = \arg\min_\theta L^-(\mathcal{D}^*, Seg; \theta; \epsilon, \zeta)$$

*The approximation of $\hat{\theta}_{\epsilon,\zeta}(-\mathcal{D}^*)$ is derived as*

$$\begin{aligned} \hat{\theta}_{\epsilon,\zeta}(-\mathcal{D}^*) &\approx \theta_{\epsilon,\zeta}(-\mathcal{D}^*) \\ &\triangleq \hat{\theta} - H_{\hat{\theta}}^{-1} \cdot \left( \epsilon \cdot \nabla_\theta Pos(\mathcal{D}^*, Seg; \hat{\theta}) + (\zeta - 1) \cdot \nabla_\theta Neg(\mathcal{D}^*, Seg; \hat{\theta}) \right) \end{aligned} \tag{17}$$

*Proof.* Assume that influence as positive sample and as negative sample can be linearly superposed. Then when the influence of dataset $\mathcal{D}^*$ as positive sample is up-weighted by $\epsilon$ and that as negative sample is up-weighted by $\zeta$, then the loss function become

$$L^-(\mathcal{D}^*, \text{Seg}; \theta; \epsilon, \zeta) \triangleq L_{Total}(\mathcal{B}; \theta) + \epsilon \cdot \text{Pos}(\mathcal{D}^*, \text{Seg}; \hat{\theta}) + (\zeta - 1) \cdot \text{Neg}(\mathcal{D}^*, \text{Seg}; \hat{\theta})$$

And corresponding parameters $\theta_{\epsilon,\zeta}$ are defined as

$$\hat{\theta}_{\epsilon,\zeta}(-\mathcal{D}^*) = \arg\min_\theta L^-(\mathcal{D}^*, \text{Seg}; \theta; \epsilon, \zeta)$$

Then, from the minimizing condition,

$$\nabla_\theta L_{Total}(\mathcal{B}; \hat{\theta}_{\epsilon,\zeta}) + \epsilon \cdot \nabla_\theta \text{Pos}(\mathcal{D}^*, \text{Seg}; \hat{\theta}_{\epsilon,\zeta}) + (\zeta - 1) \cdot \nabla_\theta \text{Neg}(\mathcal{D}^*, \text{Seg}; \hat{\theta}_{\epsilon,\zeta}) = 0,$$

where $\hat{\theta}_{\epsilon,\zeta}(-\mathcal{D}^*)$ is written as $\hat{\theta}_{\epsilon,\zeta}$ for short. Perform a Taylor expansion around $\theta = \hat{\theta}$, then we have

$$\nabla_\theta L_{Total}(\mathcal{B}; \hat{\theta}) + \epsilon \cdot \nabla_\theta \text{Pos}(\mathcal{D}^*, \text{Seg}; \hat{\theta}) + (\zeta - 1) \cdot \nabla_\theta \text{Neg}(\mathcal{D}^*, \text{Seg}; \hat{\theta})$$
$$+ \nabla_\theta^2 L_{Total}(\mathcal{B}; \hat{\theta}) \cdot \left( \hat{\theta}_{\epsilon,\zeta} - \hat{\theta} \right) = 0.$$

Because $\hat{\theta}$ minimizes $L_{Total}(\mathcal{B}; \theta)$, the first term in above equation equals $0$. Then we have

$$\hat{\theta}_{\epsilon,\zeta} \approx \hat{\theta} - H_{\hat{\theta}}^{-1} \cdot \left( \epsilon \cdot \nabla_\theta \text{Pos}(\mathcal{D}^*, \text{Seg}; \hat{\theta}) + (\zeta - 1) \cdot \nabla_\theta \text{Neg}(\mathcal{D}^*, \text{Seg}; \hat{\theta}) \right)$$

$$= \hat{\theta} - \epsilon \cdot H_{\hat{\theta}}^{-1} \cdot \nabla_\theta \text{Pos}(\mathcal{D}^*, \text{Seg}; \hat{\theta}) - (\zeta - 1) \cdot H_{\hat{\theta}}^{-1} \cdot \nabla_\theta \text{Neg}(\mathcal{D}^*, \text{Seg}; \hat{\theta})$$

$$= \hat{\theta} + \epsilon \cdot \text{positive-IF}(\mathcal{D}^*, \text{Seg}; \hat{\theta}) + (\zeta - 1) \cdot \text{negative-IF}(\mathcal{D}^*, \text{Seg}; \hat{\theta})$$

where $H_{\hat{\theta}} = \nabla_\theta^2 \sum_{(U,V) \in \mathcal{B}} L_{\text{Batch}}(U, V; \hat{\theta}) + \delta I$. When $\epsilon = -1$, $\zeta = 0$, $\hat{\theta}_{-1,0}$ estimates the parameters obtained by retraining after $\mathcal{D}^*$ removed.

$\square$

**Assumption D.3.** The loss $L_{\text{Batch}}(U, V, \theta)$ is convex and twice-differentiable in $\theta$, with positive regularization $\delta > 0$. There exists $C_H \in \mathbb{R}$ such that

$$\|\nabla_\theta^2 L_{\text{Batch}}(U, V; \theta_1) - \nabla_\theta^2 L_{\text{Batch}}(U, V; \theta_2)\|_2 \le C_H \|\theta_1 - \theta_2\|_2$$

for all $(U, V) \in \mathcal{B}$ and $\theta_1, \theta_2 \in \Theta$.

**Assumption D.4.** The function $L'((x^T, x^I); \theta)$:

$$L'((x^T, x^I); \theta) = \text{Pos}((x^T, x^I); \theta) + \text{Neg}((x^T, x^I); \theta)$$

is convex and twice-differentiable in $\theta$, with some positive regularization. There exists $C'_H \in \mathbb{R}$ such that

$$\|\nabla_\theta^2 L'((x^T, x^I); \theta_1) - \nabla_\theta^2 L'((x^T, x^I); \theta_2)\|_2 \le C'_H \|\theta_1 - \theta_2\|_2$$

for all $(x^T, x^I) \in \mathcal{D}^*$ and $\theta_1, \theta_2 \in \Theta$.

**Corollary D.5.**

$$\|\nabla_\theta^2 L^-(\mathcal{D}^*, Seg; \theta_1) - \nabla_\theta^2 L^-(\mathcal{D}^*, Seg; \theta_2)\|_2 \le (|\mathcal{B}| \cdot C_H + |\mathcal{D}^*| \cdot C'_H|) \|\theta_1 - \theta_2\|$$

*Define* $C_H^- \triangleq |\mathcal{B}| \cdot C_H + |\mathcal{D}^*| \cdot C'_H$

**Definition D.6.** Define $|\mathcal{D}|$ as the number of pairs

$$C'_L = \max_{(x^T, x^I) \in \mathcal{B}} \left\| \nabla_\theta L'((x^T, x^I); \hat{\theta}) \right\|_2,$$

$$\sigma'_{\min} = \text{smallest singular value of } \nabla_\theta^2 L^-(\mathcal{D}^*, \text{Seg}; \hat{\theta}),$$

$$\sigma_{\min} = \text{smallest singular value of } \nabla_\theta^2 L_{\text{Total}}(\mathcal{B}; \hat{\theta}),$$

Based on above corollaries and assumptions, we derive the following theorem.

**Theorem D.7.** *We obtain the error between the actual influence and our predicted influence as follows:*

$$\left\| \hat{\theta}(-\mathcal{D}^*) - \theta_{if}(-\mathcal{D}^*) \right\|$$

$$\le \frac{C'_H C_H^- |\mathcal{D}^*|^2 {C'_L}^2}{2(\sigma'_{\min} + \delta)^3} + \left| \frac{2\delta + \sigma_{\min} + \sigma'_{\min}}{(\delta + \sigma'_{\min}) \cdot (\delta + \sigma_{\min})} \right| \cdot C'_L |\mathcal{D}^*|$$

*Proof.* We will use the one-step Newton approximation as an intermediate step. Define $\Delta \theta_{Nt}(-\mathcal{D}^*)$ as

$$\Delta \theta_{Nt}(-\mathcal{D}^*) \triangleq H_\delta^{-1} \cdot \nabla_\theta L'(\mathcal{D}^*, \text{Seg}; \hat{\theta}),$$

where $H_\delta = \delta \cdot I + \nabla_\theta^2 L^-(\mathcal{D}^*, \text{Seg}; \hat{\theta})$ is the regularized empirical Hessian at $\hat{\theta}$ but reweighed after removing the influence of $\mathcal{D}^*$. Then the one-step Newton approximation for $\hat{\theta}(-\mathcal{D}^*)$ is defined as $\theta_{Nt}(-\mathcal{D}^*) \triangleq \Delta \theta_{Nt}(-\mathcal{D}^*) + \hat{\theta}$.

In the following, we will separate the error between $\theta_{if}(-\mathcal{D}^*)$ and $\hat{\theta}(-\mathcal{D}^*)$ into the following two parts:

$$\hat{\theta}(-\mathcal{D}^*) - \theta_{if}(-\mathcal{D}^*) = \underbrace{\hat{\theta}(-\mathcal{D}^*) - \theta_{Nt}(-\mathcal{D}^*)}_{\text{Err}_{\text{Nt, act}}(-\mathcal{D}^*)} + \underbrace{\left(\theta_{Nt}(-\mathcal{D}^*) - \hat{\theta}\right) - \left(\theta_{if}(-\mathcal{D}^*) - \hat{\theta}\right)}_{\text{Err}_{\text{Nt, if}}(-\mathcal{D}^*)}$$

Firstly, in **Step** 1, we will derive the bound for Newton-actual error $\text{Err}_{\text{Nt, act}}(-\mathcal{D}^*)$. Since $L^-(\theta)$ is strongly convex with parameter $\sigma'_{\min} + \delta$ and minimized by $\hat{\theta}(-\mathcal{D}^*)$, we can bound the distance $\left\|\hat{\theta}(-\mathcal{D}^*) - \theta_{Nt}(-\mathcal{D}^*)\right\|_2$ in terms of the norm of the gradient at $\theta_{Nt}$:

$$\left\|\hat{\theta}(-\mathcal{D}^*) - \theta_{Nt}(-\mathcal{D}^*)\right\|_2 \leq \frac{2}{\sigma'_{\min} + \delta} \left\|\nabla_\theta L^-(\theta_{Nt}(-\mathcal{D}^*))\right\|_2 \tag{18}$$

Therefore, the problem reduces to bounding $\|\nabla_\theta L^-(\theta_{Nt}(-\mathcal{D}^*))\|_2$. Noting that $\nabla_\theta L'(\hat{\theta}) = -\nabla_\theta L^-$. This is because $\hat{\theta}$ minimizes $L^- + L'$, that is,

$$\nabla_\theta L^-(\hat{\theta}) + \nabla_\theta L'(\hat{\theta}) = 0.$$

Recall that $\Delta\theta_{Nt} = H_\delta^{-1} \cdot \nabla_\theta L'(\mathcal{D}^*, \text{Seg}; \hat{\theta}) = -H_\delta^{-1} \cdot \nabla_\theta L^-(\mathcal{D}^*, \text{Seg}; \hat{\theta})$. Given the above conditions, we can have this bound for $\text{Err}_{\text{Nt, act}}(-\mathcal{D}^*)$.

$$
\begin{aligned}
&\left\|\nabla_\theta L^-(\theta_{Nt}(-\mathcal{D}^*))\right\|_2 \\
&= \left\|\nabla_\theta L^-\left(\hat{\theta} + \Delta\theta_{Nt}(-\mathcal{D}^*)\right)\right\|_2 \\
&= \left\|\nabla_\theta L^-\left(\hat{\theta} + \Delta\theta_{N_t}(-\mathcal{D}^*)\right) - \nabla_\theta L^-\left(\hat{\theta}\right) - \nabla_\theta^2 L^-\left(\hat{\theta}\right) \cdot \Delta\theta_{N_t}(-\mathcal{D}^*)\right\|_2 \\
&= \left\|\int_0^1 \left(\nabla_\theta^2 L^-\left(\hat{\theta} + t \cdot \Delta\theta_{Nt}(-\mathcal{D}^*)\right) - \nabla_\theta^2 L^-\left(\hat{\theta}\right)\right) \Delta\theta_{Nt}(-\mathcal{D}^*)\, dt\right\|_2 \\
&\leq \frac{C_H^-}{2} \|\Delta\theta_{Nt}(-\mathcal{D}^*)\|_2^2 = \frac{C_H^-}{2} \left\|\left[\nabla_\theta^2 L^-(\hat{\theta})\right]^{-1} \nabla_\theta L^-(\hat{\theta})\right\|_2^2 \\
&\leq \frac{C_H^-}{2(\sigma'_{\min} + \delta)^2} \left\|\nabla_\theta L^-(\hat{\theta})\right\|_2^2 = \frac{C_H^-}{2(\sigma'_{\min} + \delta)^2} \left\|\nabla_\theta L'(\hat{\theta})\right\|_2^2 \\
&\leq \frac{C_H^- \|\mathcal{D}^*\|^2 C_L'^2}{2(\sigma'_{\min} + \delta)^2}.
\end{aligned}
\tag{19}
$$

Now we come to **Step** 2 to bound $\text{Err}_{\text{Nt, if}}(-\mathcal{D}^*)$, and we will bound the difference in parameter change between Newton and our ECIF method.

$$
\begin{aligned}
&\left\|\left(\theta_{Nt}(-\mathcal{D}^*) - \hat{\theta}\right) - \left(\theta_{if}(-\mathcal{D}^*) - \hat{\theta}\right)\right\| \\
&= \left\|\left[\left(\delta \cdot I + \nabla_\theta^2 L^-\left(\hat{\theta}\right)\right)^{-1} + \left(\delta \cdot I + \nabla_\theta^2 L_{\text{Total}}\left(\hat{\theta}\right)\right)^{-1}\right] \cdot \nabla_\theta L'(\mathcal{D}^*, \text{Seg}; \hat{\theta})\right\|
\end{aligned}
$$

For simplification, we use matrix $A$, $B$ for the following substitutions:

$$A = \delta \cdot I + \nabla_\theta^2 L^-\left(\hat{\theta}\right)$$

$$B = \delta \cdot I + \nabla_\theta^2 L_{\text{Total}}\left(\hat{\theta}\right)$$

And $A$ and $B$ are positive definite matrices with the following properties

$$\delta + \sigma'_{\min} \prec A \prec \delta + \sigma'_{\max}$$
$$\delta + \sigma_{\min} \prec B \prec \delta + \sigma_{\max}$$

Therefore, we have

$$
\begin{aligned}
& \left\| \left( \theta_{Nt}(-\mathcal{D}^*) - \hat{\theta} \right) - \left( \theta_{if}(-\mathcal{D}^*) - \hat{\theta} \right) \right\| \\
& = \left\| \left( A^{-1} + B^{-1} \right) \cdot \nabla_\theta L^-(\mathcal{D}^*, \mathrm{Seg}; \hat{\theta}) \right\| \\
& \leq \left\| A^{-1} + B^{-1} \right\| \cdot \left\| \nabla_\theta L^-(\mathcal{D}^*, \mathrm{Seg}; \hat{\theta}) \right\| \\
& \leq \left| \frac{2\delta + \sigma_{\min} + \sigma'_{\min}}{(\delta + \sigma'_{\min}) \cdot (\delta + \sigma_{\min})} \right| \cdot \left\| \nabla_\theta L^-(\mathcal{D}^*, \mathrm{Seg}; \hat{\theta}) \right\| \\
& \leq \left| \frac{2\delta + \sigma_{\min} + \sigma'_{\min}}{(\delta + \sigma'_{\min}) \cdot (\delta + \sigma_{\min})} \right| \cdot C'_L |\mathcal{D}^*|
\end{aligned}
\tag{20}
$$

By combining the conclusions from Step I and Step II in Equations 18, 19 and 20, we obtain the error between the actual influence and our predicted influence as follows:

$$
\begin{aligned}
& \left\| \hat{\theta}(-\mathcal{D}^*) - \theta_{if}(-\mathcal{D}^*) \right\| \\
& \leq \frac{C'_H C_H^- |\mathcal{D}^*|^2 {C'_L}^2}{2(\sigma'_{\min} + \delta)^3} + \left| \frac{2\delta + \sigma_{\min} + \sigma'_{\min}}{(\delta + \sigma'_{\min}) \cdot (\delta + \sigma_{\min})} \right| \cdot C'_L |\mathcal{D}^*|.
\end{aligned}
$$

It is notable that such error bound is small when the number of removal samples $|\mathcal{D}^*|$ is fixed as in practice $\delta = O(|\mathcal{B}|)$. $\qquad \square$

# E APPLICATIONS OF ECIF

## E.1 TASK-RELATED IS

**Property E.1.** *Considering a specific set $\mathcal{D}'$ with text and image embeddings $U'$ and $V'$, and a dataset $D^*$ to be removed, then we have*

$$
\begin{aligned}
& L_{Batch}(U', V'; \hat{\theta}(-D^*)) - L_{Batch}(U', V'; \hat{\theta}) \approx \nabla L_{Batch}(U', V'; \hat{\theta})^{\mathrm{T}}(\hat{\theta}(-D^*) - \hat{\theta}) \\
& = -\nabla L_{Batch}(U', V'; \hat{\theta})^{\mathrm{T}} \cdot \left( \text{positive-IF}(\mathcal{D}^*, \mathrm{Seg}; \hat{\theta}) + \text{negative-IF}(\mathcal{D}^*, \mathrm{Seg}; \hat{\theta}) \right).
\end{aligned}
\tag{21}
$$

*where $\hat{\theta}(-D^*)$ is the optimal model for the loss eliminating $D^*$, positive-IF$(\mathcal{D}^*; Seg; \hat{\theta})$ and negative-IF$(\mathcal{D}^*, Seg; \hat{\theta})$ are obtained from Proposition 4.3 for $D^*$.*

*Proof.*

$$
\begin{aligned}
\mathrm{IS}(\mathcal{D}', \mathcal{D}^*; \mathrm{Seg}) & \triangleq - \left. \frac{\mathrm{d}L_{\mathrm{Batch}}(U', V'; \theta_{\epsilon, \zeta=0})}{\mathrm{d}\epsilon} \right|_{\epsilon=0} - \left. \frac{\mathrm{d}L_{\mathrm{Batch}}(U', V'; \theta_{\epsilon=0, \zeta})}{\mathrm{d}\zeta} \right|_{\zeta=0} \\
& \approx - \nabla L_{\mathrm{Batch}}(U', V'; \hat{\theta})^{\mathrm{T}} \cdot \left( \text{positive-IF}(\mathcal{D}^*, \mathrm{Seg}; \hat{\theta}) + \text{negative-IF}(\mathcal{D}^*, \mathrm{Seg}; \hat{\theta}) \right)
\end{aligned}
$$

$\qquad \square$

## E.2 RELATIVE INFLUENCE SCORE

**Proposition E.2.** *Define $I = [\text{positive-IF}(x; \hat{\theta}), \text{negative-IF}(x; \hat{\theta})]$. If the $2 \times 2$ matrix $I^{\mathrm{T}} \cdot I$ is irreversible, then the optimization problem*

$$
\arg \max_{x \in \mathcal{D}} \max_{\epsilon, \zeta} \left| L_{Batch}(U', V'; \hat{\theta} + \Delta\hat{\theta}_{\epsilon, \zeta}(x)) - L_{Batch}(U', V'; \hat{\theta}) \right| \quad s.t. \left\| \Delta\hat{\theta}_{\epsilon, \zeta}(x) \right\|^2 \leq \delta^2 \tag{22}
$$

*is equivalent to*

$$
\arg \max_{x \in \mathcal{D}} \| \text{negative-IF}(x; \hat{\theta}) \|_2^{-1} \left| \nabla L_{Batch}(U', V'; \hat{\theta})^{\mathrm{T}} \cdot \text{negative-IF}(x; \hat{\theta}) \right|.
$$

*Else, $I^{\mathrm{T}} \cdot I$ is reversible, then the initial problem is equivalent to*

$$
\arg \max_{x \in \mathcal{D}} \| \nabla L_{Batch}(U', V'; \hat{\theta}) \|_2^{-1} \left| \nabla L_{Batch}(U', V'; \hat{\theta})^{\mathrm{T}} \cdot I \cdot \left[ I^{\mathrm{T}} \cdot I \right]^{-1} \cdot I^{\mathrm{T}} \cdot \nabla L_{Batch}(U', V'; \hat{\theta}) \right|.
$$

*Proof.* From (17), we have

$$\left| L_{\text{Batch}}(U', V'; \hat{\theta} + \Delta\hat{\theta}_{\epsilon,\zeta}(x)) - L_{\text{Batch}}(U', V'; \hat{\theta}) \right| \tag{23}$$

$$\approx \left| \nabla L((U', V'); \hat{\theta})^{\text{T}} \cdot \Delta\hat{\theta}_{\epsilon,\zeta}(x)) \right| \tag{24}$$

$$\approx \left| \nabla L((U', V'); \hat{\theta})^{\text{T}} \cdot \left( \epsilon \cdot \text{positive-IF}((x^T, x^I); \hat{\theta}) + (\zeta - 1) \, \text{negative-IF}((x^T, x^I); \hat{\theta}) \right) \right| \tag{25}$$

And still from (17), the constraint in parameter changes can be written as

$$\left\| \Delta\hat{\theta}_{\epsilon,\zeta}(x) \right\| \tag{26}$$

$$= \left\| \epsilon \cdot \text{positive-IF}((x^T, x^I); \hat{\theta}) + (\zeta - 1) \, \text{negative-IF}((x^T, x^I); \hat{\theta}) \right\| \leq \delta \tag{27}$$

We can regard (25) as the inner product between vector $u \triangleq \nabla L((U', V'); \hat{\theta})$ and vector $v \triangleq \epsilon \cdot \text{positive-IF} + (\zeta - 1) \, \text{negative-IF}$.

If positive-IF is not parallel to negative-IF, then the constraint in equation (26) becomes that vector $v$ is chosen from a ball of radius $\delta$. Otherwise, the constraint is equivalent to a constraint on the norm of a vector that is parallel to positive-IF or negative-IF. Therefore, we will proceed with a classification discussion based on whether positive-IF and negative-IF are parallel.

Firstly, we consider the $\parallel$ case. As is well known, the inner product of vectors reaches its extreme when the two vectors are parallel. We can choose $\epsilon$ and $\zeta$ freely to make vectors $v \parallel u$. Assume that there exists $c \in \mathbb{R}$ s.t.

$$[\text{positive-IF}, \text{negative-IF}] \cdot \begin{bmatrix} \epsilon \\ \zeta - 1 \end{bmatrix} = c \cdot \nabla L((U', V'); \hat{\theta})$$

Denote $[\text{positive-IF}, \text{negative-IF}]$ as $I$

$$[\text{positive-IF}, \text{negative-IF}] \cdot \begin{bmatrix} \epsilon \\ \zeta - 1 \end{bmatrix} = c \cdot \nabla L((U', V'); \hat{\theta})$$

$$\begin{bmatrix} \text{positive-IF}^{\text{T}} \\ \text{negative-IF}^{\text{T}} \end{bmatrix} \cdot [\text{positive-IF}, \text{negative-IF}] \cdot \begin{bmatrix} \epsilon \\ \zeta - 1 \end{bmatrix} = c \cdot \begin{bmatrix} \text{positive-IF}^{\text{T}} \\ \text{negative-IF}^{\text{T}} \end{bmatrix} \cdot \nabla L((U', V'); \hat{\theta})$$

$$I^{\text{T}} \cdot I \cdot \begin{bmatrix} \epsilon \\ \zeta - 1 \end{bmatrix} = c \cdot I^{\text{T}} \cdot \nabla L((U', V'); \hat{\theta})$$

$$\begin{bmatrix} \epsilon \\ \zeta - 1 \end{bmatrix} = c \cdot \left[ I^{\text{T}} \cdot I \right]^{-1} \cdot I^{\text{T}} \cdot \nabla L((U', V'); \hat{\theta})$$

Noting that $I^{\text{T}} \cdot I$ is invertible matrix as long as positive-IF, negative-IF are not parallel. Considering the constraints of the length of vector $v$, then

$$\| c \cdot \nabla L((U', V'); \hat{\theta}) \| \leq \delta$$

We can make vector $v$ reach its largest norm with setting $c$ to an appropriate number:

$$c = \frac{\delta}{\| \nabla L((U', V'); \hat{\theta}) \|}$$

Finally, we obtain the expression of vector 2 that maximizes expression (23)

$$[\text{positive-IF}, \text{negative-IF}] \cdot \begin{bmatrix} \epsilon \\ \zeta - 1 \end{bmatrix} = c \cdot I \cdot \left[ I^{\text{T}} \cdot I \right]^{-1} \cdot I^{\text{T}} \cdot \nabla L((U', V'); \hat{\theta})$$

Then we have

$$\left| L((U', V'); \theta_{\epsilon,\zeta}(x^T, x^I)) - L((U', V'); \hat{\theta}) \right|$$

$$= \left| \nabla L((U', V'); \hat{\theta})^{\text{T}} \cdot \left( [\text{positive-IF}, \text{negative-IF}] \cdot \begin{bmatrix} \epsilon \\ \zeta - 1 \end{bmatrix} \right) \right|$$

$$= c \cdot \left| \nabla L((U', V'); \hat{\theta})^{\text{T}} \cdot I \cdot \left[ I^{\text{T}} \cdot I \right]^{-1} \cdot I^{\text{T}} \cdot \nabla L((U', V'); \hat{\theta}) \right|$$

where $I = [\text{positive-IF}, \text{negative-IF}]$.

$$\arg\max_{(x^T, x^I) \in \mathcal{D}} \frac{\delta}{\|\nabla L((U', V'); \hat{\theta})\|} \cdot \left| \nabla L((U', V'); \hat{\theta})^{\mathrm{T}} \cdot I \cdot \left[ I^{\mathrm{T}} \cdot I \right]^{-1} \cdot I^{\mathrm{T}} \cdot \nabla L((U', V'); \hat{\theta}) \right|$$

where $I = \left[ \text{positive-IF}((x^T, x^I); \hat{\theta}), \text{negative-IF}((x^T, x^I); \hat{\theta}) \right]$.

If positive-IF, negative-IF are not parallel, the optimization problem in form (22) is equivalent to

$$\arg\max_{x \in \mathcal{D}} \frac{\delta}{\|\nabla L((U', V'); \hat{\theta})\|} \left| \nabla L((U', V'); \hat{\theta})^{\mathrm{T}} I \left[ I^{\mathrm{T}} \cdot I \right]^{-1} I^{\mathrm{T}} \nabla L((U', V'); \hat{\theta}) \right|.$$

Because $\delta$ is independent of data, we can drop it and write the above equation as

$$\arg\max_{x \in \mathcal{D}} \|\nabla L((U', V'); \hat{\theta})\|^{-1} \left| \nabla L((U', V'); \hat{\theta})^{\mathrm{T}} I \left[ I^{\mathrm{T}} \cdot I \right]^{-1} I^{\mathrm{T}} \nabla L((U', V'); \hat{\theta}) \right|.$$

Then, we come to the second case where positive-IF $\|$ negative-IF. We can define a

$$\left\| \Delta \hat{\theta}_{\epsilon, \zeta}(x) \right\| \tag{28}$$

$$= \left\| \epsilon \cdot \text{positive-IF}((x^T, x^I); \hat{\theta}) + (\zeta - 1) \, \text{negative-IF}((x^T, x^I); \hat{\theta}) \right\| \tag{29}$$

$$\triangleq \left\| \alpha(\epsilon, \zeta) \cdot \text{positive-IF}((x^T, x^I); \hat{\theta}) \right\| \leq \delta \tag{30}$$

And the constraint is imposed on $\alpha$ by

$$\alpha(\epsilon, \zeta) \leq \frac{\delta}{\|\text{positive-IF}((x^T, x^I); \hat{\theta})\|}$$

Therefore, equation (23) is equivalent to

$$\max_{\epsilon, \zeta} \left| \nabla L((U', V'); \hat{\theta})^{\mathrm{T}} \cdot \left( \alpha(\epsilon, \zeta) \cdot \text{positive-IF}((x^T, x^I); \hat{\theta}) \right) \right|$$

$$= \max_{\epsilon, \zeta} \alpha(\epsilon, \zeta) \cdot \left| \nabla L((U', V'); \hat{\theta})^{\mathrm{T}} \cdot \left( \text{positive-IF}((x^T, x^I); \hat{\theta}) \right) \right|$$

$$= \frac{\delta}{\|\text{positive-IF}((x^T, x^I); \hat{\theta}))\|} \cdot \left| \nabla L((U', V'); \hat{\theta})^{\mathrm{T}} \cdot \text{positive-IF}((x^T, x^I); \hat{\theta}) \right|$$

$$= \frac{\delta}{\|\text{negative-IF}((x^T, x^I); \hat{\theta})\|} \cdot \left| \nabla L((U', V'); \hat{\theta})^{\mathrm{T}} \cdot \text{negative-IF}((x^T, x^I); \hat{\theta}) \right|$$

Because $\delta$ is independent of data, we can drop it and write the above equation as

$$\|\text{positive-IF}((x^T, x^I); \hat{\theta}))\|^{-1} \cdot \left| \nabla L((U', V'); \hat{\theta})^{\mathrm{T}} \cdot \text{positive-IF}((x^T, x^I); \hat{\theta}) \right|$$

$$= \|\text{negative-IF}((x^T, x^I); \hat{\theta})\|^{-1} \cdot \left| \nabla L((U', V'); \hat{\theta})^{\mathrm{T}} \cdot \text{negative-IF}((x^T, x^I); \hat{\theta}) \right|.$$

$\square$

# F ADDITIONAL EXPERIMENTAL RESULTS

## F.1 DETAILS OF EXPERIMENT SETTINGS

**Datasets.** We employ three datasets for our utility and efficiency evaluation tasks, as well as for the misprediction traceback experiments: *FGVC-Aircraft dataset* (Maji et al., 2013), *Food101 dataset* (Bossard et al., 2014), *Flowers102 dataset* (Nilsback & Zisserman, 2008). The FGVC-Aircraft dataset comprises 10,000 images of airplanes, each annotated with the model and bounding box of the dominant aircraft depicted. The Food-101 dataset, publicly available for food image recognition, includes 101 food categories, with each category containing 1,000 images. The images feature food photographs captured from various angles and under different lighting conditions. The Flowers-102 dataset consists of 102 classes of flowers native to the United Kingdom, with each

Table 4: Comparison of different removal and update strategies on CIFAR-100.

| Removal Type | Method | Accuracy (%) | Time (s) |
|---|---|---|---|
| Random | Retrain | $73.50 \pm 0.35$ | $12.56 \pm 0.37$ |
| | IF Update | $73.00 \pm 0.20$ | $7.40 \pm 0.11$ |
| Positive | Retrain | $73.50 \pm 0.41$ | $8.02 \pm 0.15$ |
| | IF Update | $72.92 \pm 0.31$ | $2.70 \pm 0.17$ |
| Negative | Retrain | $72.83 \pm 0.12$ | $7.92 \pm 0.01$ |
| | IF Update | $73.00 \pm 0.20$ | $2.26 \pm 0.19$ |

class containing between 40 and 258 images. We use *Cifar-10 dataset* (Krizhevsky, 2009) for the misalignment detection tasks.

**Implementation Details.** Our experiments utilized an Nvidia V100-32G GPU and 10 CPU cores with 64 GB memory. For all the following tasks, we employ the CLIP model 'ViT-B/16' and use LoRA few-shot learning.

For utility evaluation, when testing our method on a random sample-removing task, 10

For the experiment of *Identifying influential data for fine-tuning*, we first calculate the task-related IS for every individual sample and collect valuable data with positive IS, then choose to remove 00-30

The *multiple samples removal* experiments are conducted on Food101, Flowers102, FGVC-Aircraft, and DTD datasets, with removal ratios from $1\%$ to $7\%$, respectively.

For the *misprediction trace back* task, we conduct experiments on Food101, Flowers102, FGVC-Aircraft, and DTD datasets. We first choose a mispredicted test sample as the target in algorithm 3, then calculate the relative IS for each individual sample in the training dataset. Noting the relative IS is always positive. We visualize training samples with top-10 relative IS.

For the *misalignment detection* tasks, Cifar-10 and imagenette (smaller version of imageNet) datasets are used. We also applied standard data augmentation techniques on the training set,i.e., random cropping and random flipping. The model is optimized with Adam with weight decay ($5e - 1$), and $\beta$ is set to 0.9. A dropout ratio of 0.25 is used. The training iterations are set to 30, with a learning rate of $2e - 4$ and a batch size of 16. The rank of the low-rank matrices of LoRA is set to 2. We trained the model on a poisoned version of the dataset ($20\%$ / $30\%$ of the data samples are mislabeled). Then, we compute the influence score IS of all the training samples on the mispredicted test samples. At the end, we visualize the training samples that have the highest positive IS score.

## F.2 Extending Utility and Efficiency Evaluation to Larger Dataset

We use *Cifar-100 dataset* (Krizhevsky et al., 2009) for our utility and efficiency evaluation tasks. Results in table 4 4 highlight the performance of various removal and update strategies on CIFAR-100. The results demonstrate the effectiveness of IF Update (influence function) compared to traditional retraining in terms of both accuracy and computational efficiency. Across all removal types (Random, Positive, and Negative), IF Update consistently achieves comparable or higher accuracy while significantly reducing runtime.

For Random data removal, IF Update improves accuracy from $66.67\% \pm 2.36\%$ (Retrain) to $73.33\% \pm 1.18\%$ and nearly halves the runtime, decreasing from $6.87 \pm 0.19$ seconds to $3.85 \pm 0.13$ seconds. Similarly, under Positive removal, IF Update achieves an accuracy boost from $68.33\% \pm 1.18\%$ (Retrain) to $72.50\% \pm 2.04\%$, with runtime reduced from $3.01 \pm 0.05$ seconds to $0.97 \pm 0.09$ seconds. Lastly, for Negative removal, while the retrained model yields a slightly higher accuracy ($73.33\% \pm 2.36\%$) compared to IF Update ($70.83\% \pm 1.18\%$), IF Update achieves comparable performance with a runtime of $3.49 \pm 4.16$ seconds, closely matching retraining ($3.00 \pm 0.04$ seconds).

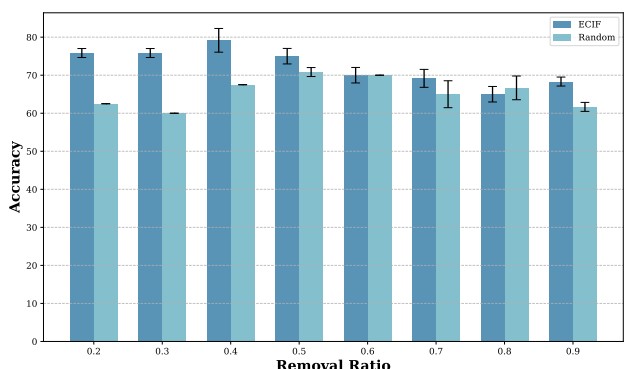

Figure 2: Harmful Data Removal on ANIMAL-10N

These results validate the utility of IF Update as a computationally efficient alternative to retraining, achieving near-equivalent or superior accuracy across varied removal scenarios on CIFAR-100.

### F.3 EXTENDING HARMFUL DATA REMOVAL TO REAL-WORLD NOISY DATASET

We use *ANIMAL-10N dataset* (Song et al., 2019) for our harmful data removal tasks.. It's a real world noisy dataset, containing five pairs of "confusing" animals: (cat, lynx), (jaguar, cheetah), (wolf, coyote), (chimpanzee, orangutan), (hamster, guinea pig), where two animals in each pair look very similar. Overall, the proportion of incorrect labels was 6.44%.

Harmful removal task on ANIMAL-10N is presented in table 2. We observe that when a portion of harmful data is removed, the ECIF method significantly outperforms random removal, particularly when the removal proportion is small. Specifically, when less than 40% of the data is removed, ECIF achieves an accuracy improvement of over 10% compared to random removal. This demonstrates the capability of ECIF to accurately identify harmful samples, thereby substantially enhancing the model's performance.

To provide a method as the reference, we adopt CLIPScore (Hessel et al., 2021), a basic data evaluation method, as the baseline for MLLM. This method is model-independent and is limited to evaluating data quality rather than assessing the contribution of the data to the model. In the task

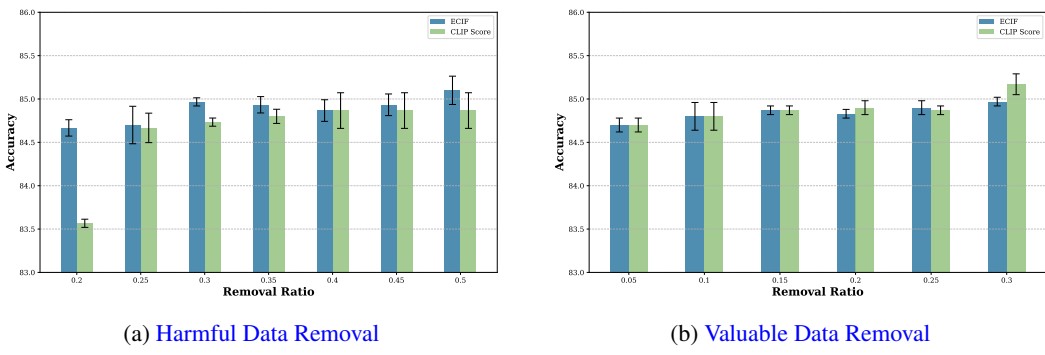

(a) Harmful Data Removal  (b) Valuable Data Removal

Figure 3: Comparison between different methods for data removal on Food101

of harmful data removal, the ECIF method demonstrates significantly better performance compared to the CLIP Score. For valuable data removal, ECIF performs slightly better than the CLIP Score. This superiority is primarily attributed to ECIF's ability to attribute data based on the relationship between the model and the data, whereas CLIP Score is solely used to evaluate data quality without considering the model's involvement.

### F.4 EVALUATING MULTIPLE SAMPLES

To comprehensively evaluate the data removal capabilities of ECIF in various scenarios, we conducted experiments on the performance when multiple samples need to be removed. Specifically, we consider the different ratios of samples (1-7%) for removal. As shown in Figure 4, we can see the accuracy difference between these two methods is very small (less than 1.5%) in most cases, except the case of 2% for Food101. These results show the utility of ECIF compared to the ground truth. Note that in Table 1, we have shown that the speed of ECIF is more than two times faster than that of retraining. Thus, ECIF is an editing method that achieves a trade-off between speed and effectiveness.

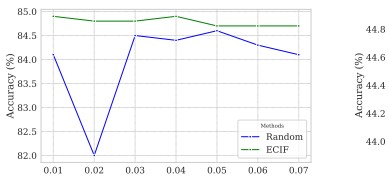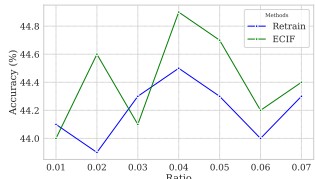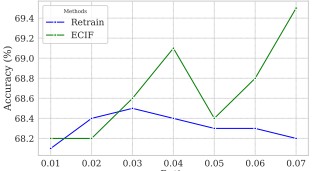

Figure 4: Impact of Remove Ratio on Food101, DTD and Flower102 datasets.

### F.5 ADDITIONAL RESULTS FOR ENHANCING FINE-TUNING VIA TASK-RELATED INFLUENCE SCORE

We demonstrate our additional results of using task-related IS to identify harmful data on Flower102 in Figure 5.

### F.6 ADDITIONAL VISUALIZATION OF MISPREDICTION TRACE BACK

We demonstrate our additional visualization results of the mispredicted data tracing in Table 5-7 and Figure 7-9.

### F.7 ADDITIONAL VISUALIZATION OF MISALIGNMENT DATA DETECTION

We demonstrate our additional results of the Visualization of the misalignment data detection in Figure 6.

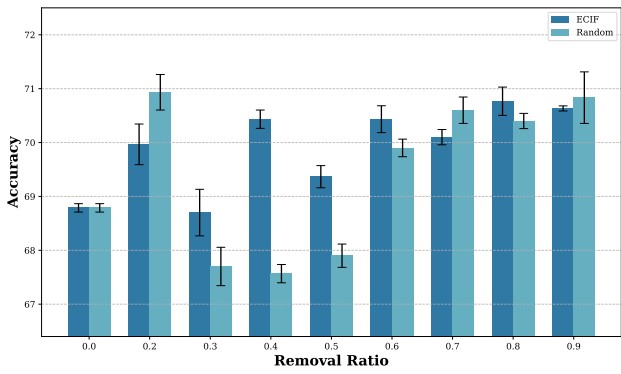

Figure 5: Harmful Data Removal on Flower102

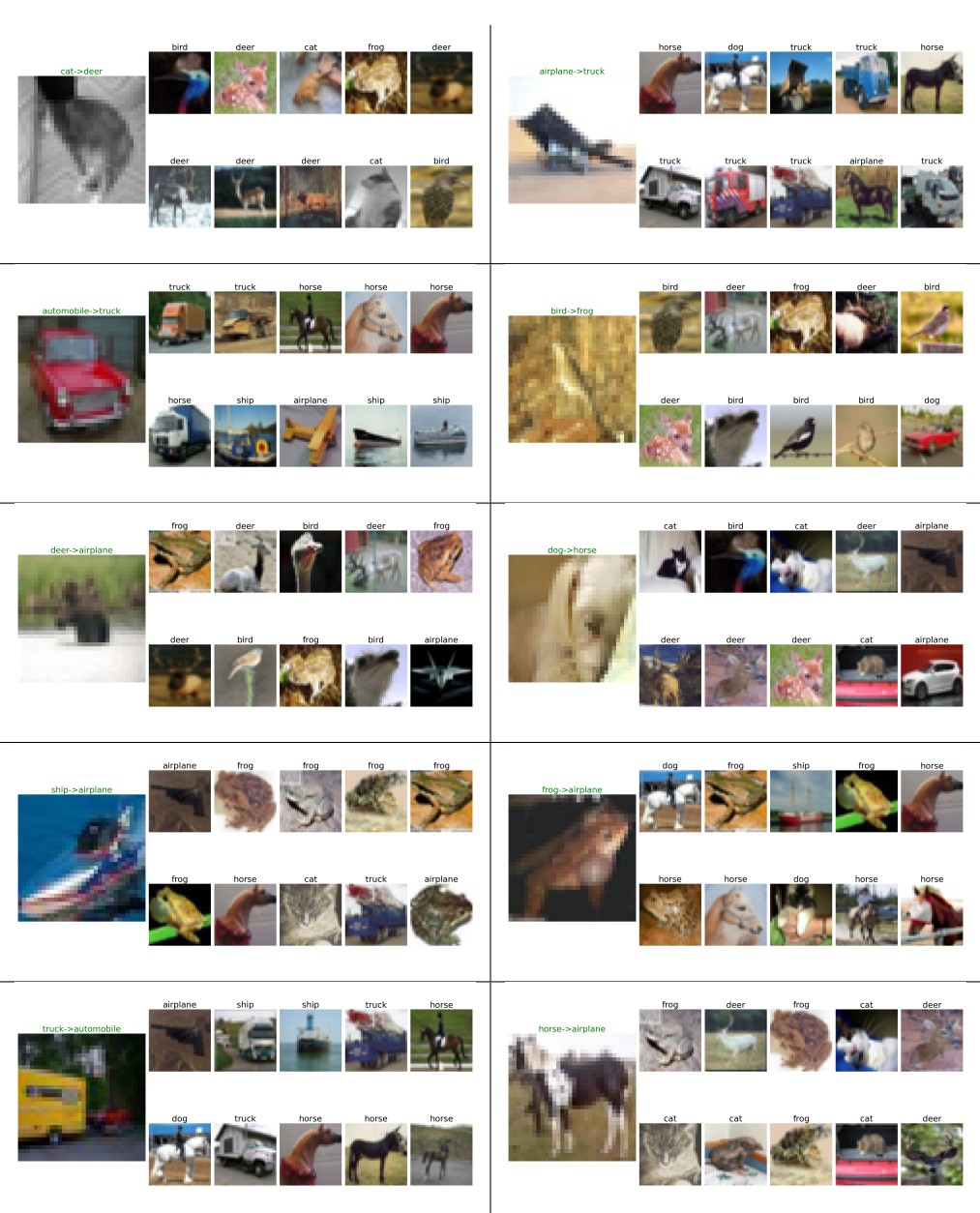

Table 5: Top-10 related test data tracing of mispredicted data on cifar-10 dataset with 10% noise data.

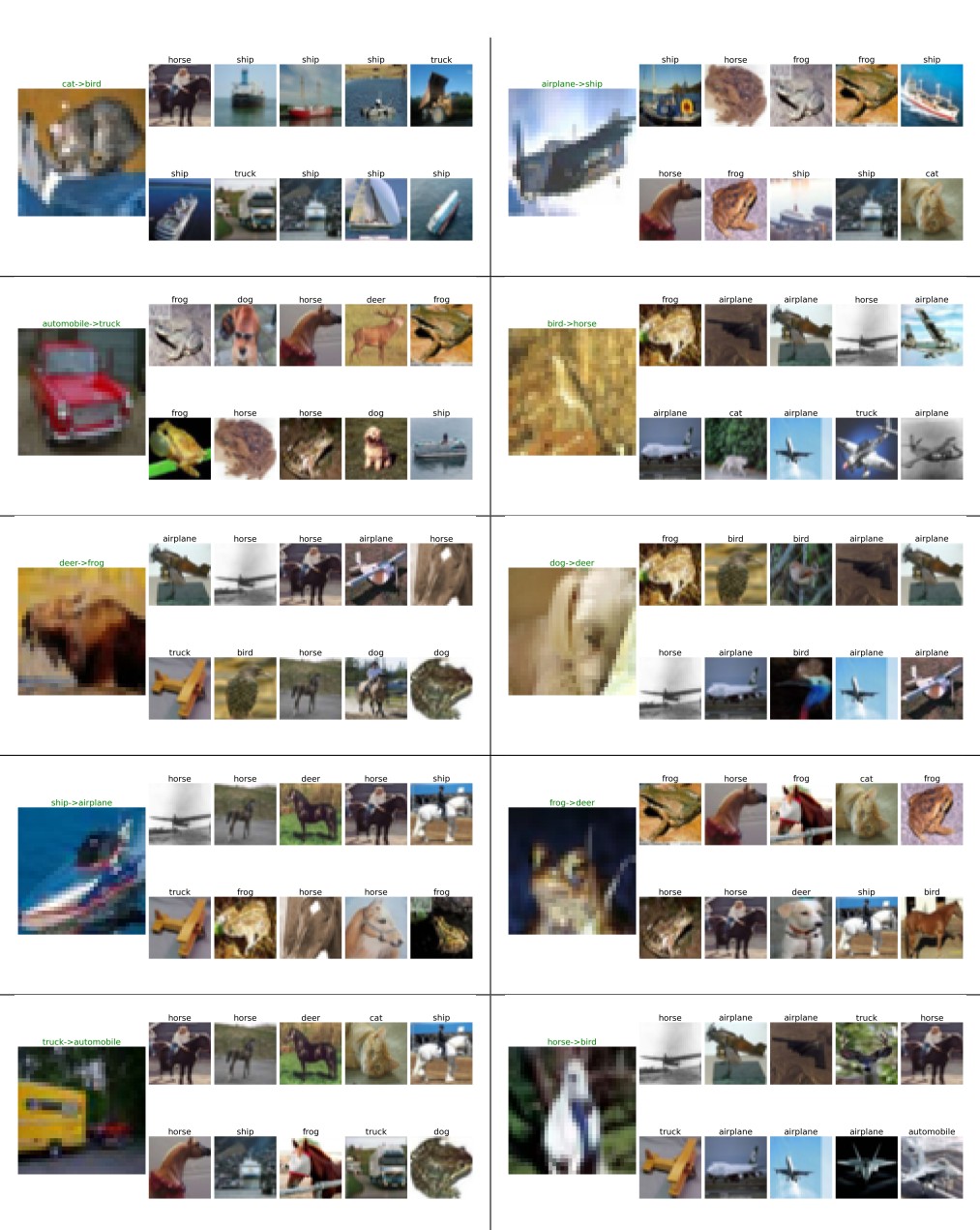

Table 6: Top-10 related test data tracing of mispredicted data on cifar-10 dataset with 20% noise data.

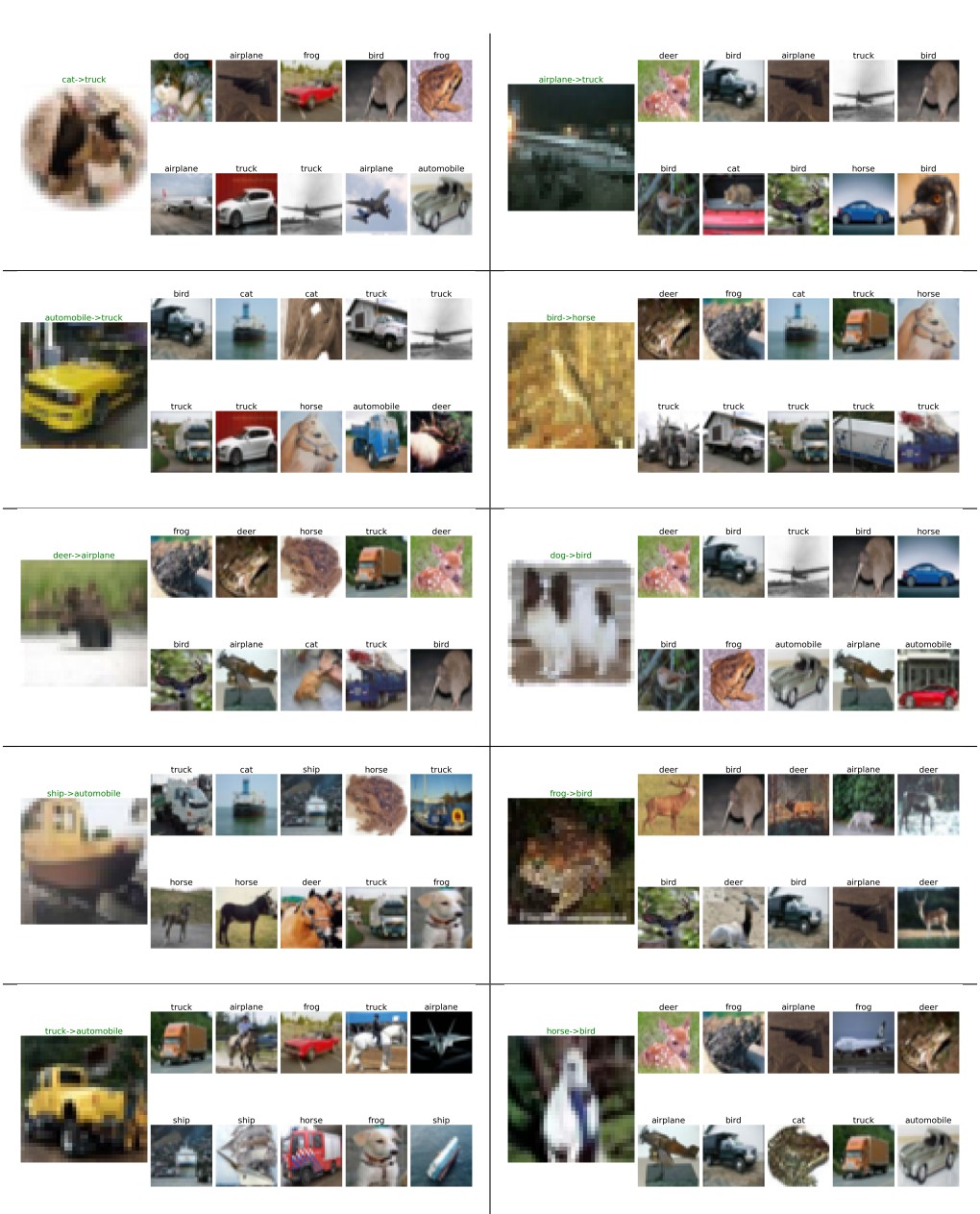

Table 7: Top-10 related test data tracing of mispredicted data on cifar-10 dataset with 30% noise data.

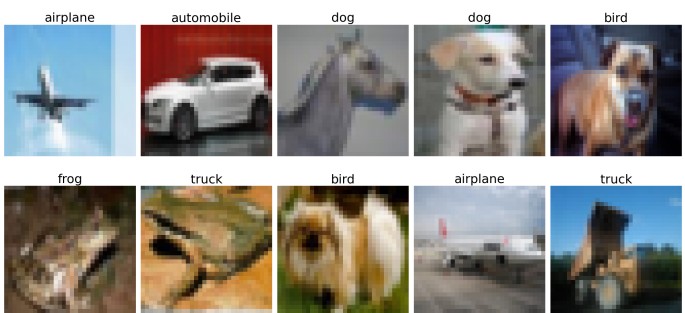

Figure 6: Visualization results for misalignment detection. 30% of the training samples were mislabeled. The figure shows the training samples that have the top-10 highest IS scores on the cifar-10 test set.

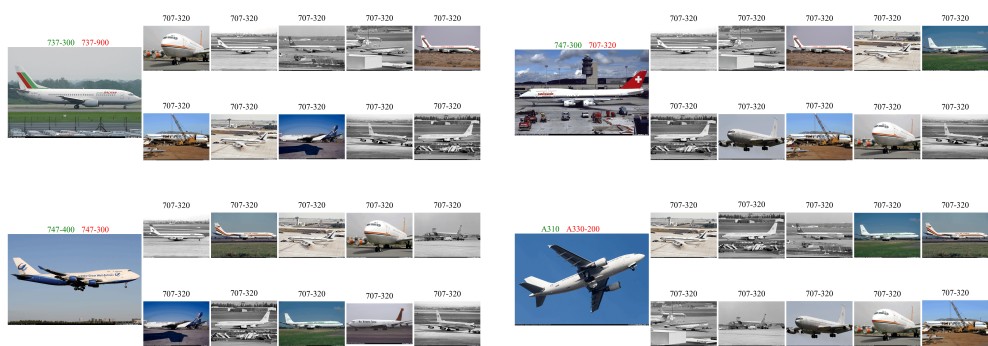

Figure 7: Top-10 related test data tracing of mispredicted data on FGVC-Aircraft with 30% noise data.

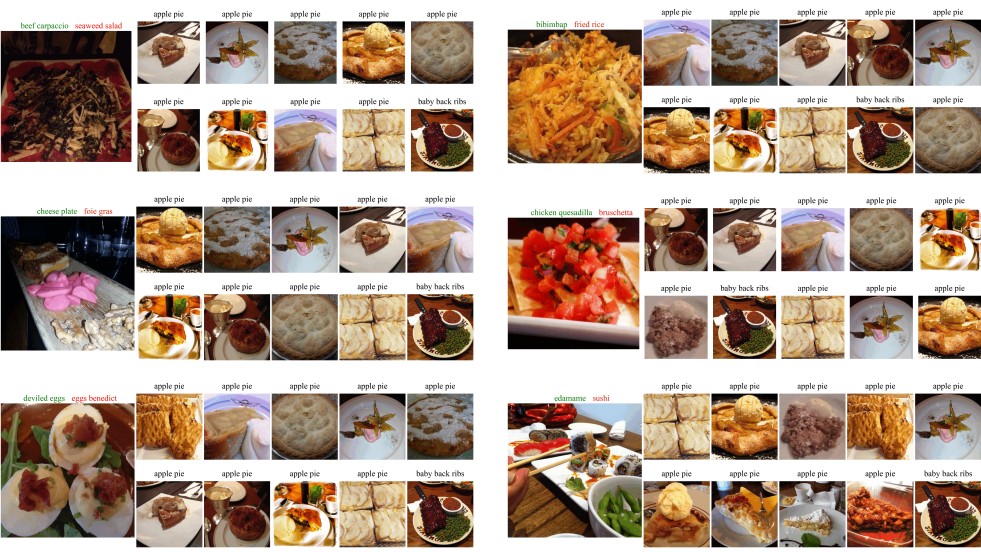

Figure 8: Top-10 related test data tracing of mispredicted data on Food-101 with 30% noise data.

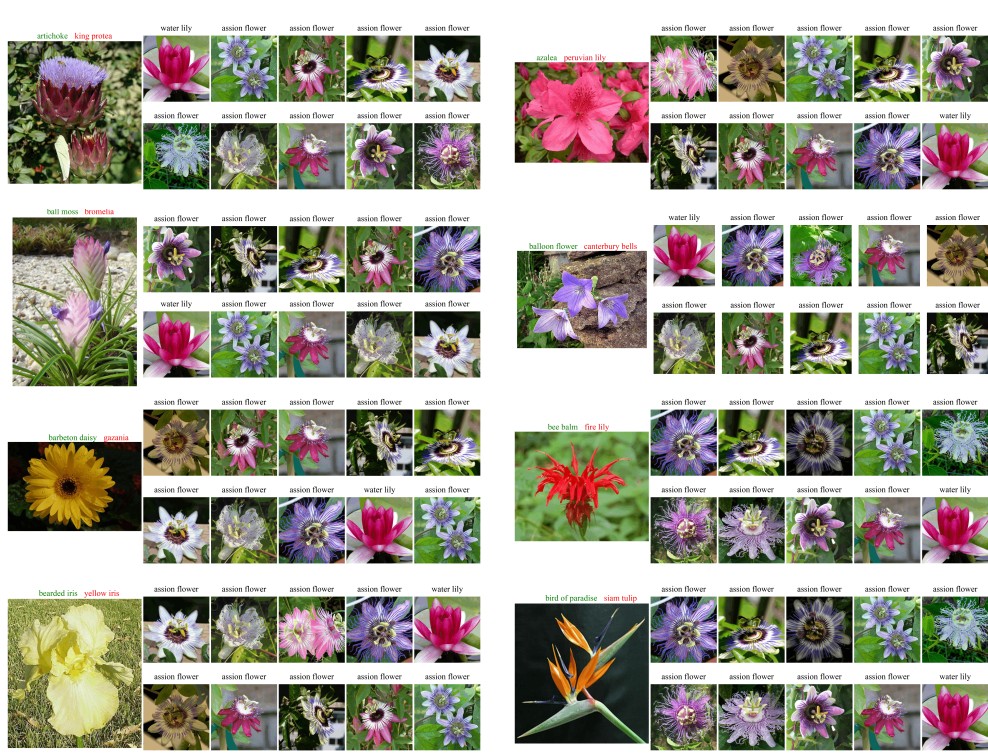

Figure 9: Top-10 related test data tracing of mispredicted data on Flowers-102 with 30% noise data.

