# OpenReview forum: "Dissecting Misalignment of Multimodal Large Language Models via Influence Function"
_ICLR.cc/2025/Conference — Submitted to ICLR 2025_

### Official Review · Reviewer_PRsw · 2024-10-27

**Soundness:** 2
**Presentation:** 2
**Contribution:** 2
**Rating:** 5
**Confidence:** 4

**Summary:**

This paper proposed extended influence function for contrastive loss. It evaluates the training data contribution in MLLMs. The proposed method tackles the challenges of applying influence functions to contrastive learning models, and propose algorithms for data evaluation, misalignment detection, and misprediction trace-back tasks. The experiments show that ECIF improves the transparency and interpretability of MLLMs.

**Strengths:**

(1) The paper addresses the misalignment of text-image pairs in training data and its impact on model performance.

(2) The proposed ECIF method provides an approach to evaluate the contribution of training data in contrastive learning models.

**Weaknesses:**

(1) The dataset used in the experiment is too simple; All of them are old and toy datasets. More complex datasets should be used, for example Imagenet instead of Imagenette.

(2) The experiments were not thorough enough, leading to the conclusions being insufficiently validated.

**Questions:**

N/A

---

> ### Author Response · Authors · 2024-11-24
> **Respone to Reviewer PRsw**
>
> ### Weaknesses:
>
> >**W1:**
> The dataset used in the experiment is too simple; All of them are old and toy datasets. More complex datasets should be used, for example Imagenet instead of Imagenette.
>
>
> We acknowledge that the datasets selected, such as Imagenette, are simpler and may not fully capture the complexities of real-world scenarios. Our initial focus was on these datasets to establish a baseline and validate the basic effect of our method.
>
> We extened our method to **Cifar100**, and the efficiency and utility results are shown here:
>
>
> | Method                     | Accuracy (%) | Time (s)     |
> | -------------------------- | ------------ | ------------ |
> | Random Removal & Retrain   | 73.50 ± 0.35 | 12.56 ± 0.37 |
> | Random Removal & IF Update | 73.00 ± 0.20 | 7.40 ± 0.11  |
>
> *Table 1: Result of the efficiency and utility experiment on Cifar 100 dataset.*
>
> From Table 1, we can find that ECIF achieves a performance comparable to that of retraining, while requiring only half the running time. This indicatesd that our method is still effective for complex large dataset.
>
> Besides, we conduct harmful removal experiments on real-world noisy dataset: ANIMAL-10. Here we use CLIPScore[1], a basic data evaluation method for MLLM as another reference method. This method is model-independent and is limited to evaluating data quality rather than assessing the contribution of the data to the model.
>
> The harmful data identified by CLIPScore correspond to those with low scores. We first remove the harmful data recognized by IF and ClipScore, then retrain the model and test the performance. The results are showed in Table 2.
>
> | Threshold | Accuracy(IF)  | Accuracy(ClipScore) |
> | --------- | ------------ | ------------------- |
> | 0.05      | 84.73 ± 0.21 | 84.90 ± 0.00        |
> | 0.10      | 84.70 ± 0.24 | 84.90 ± 0.00        |
> | 0.15      | 84.80 ± 0.29 | 84.90 ± 0.00        |
> | 0.20      | 84.67 ± 0.09 | 83.57 ± 0.05        |
> | 0.25      | 84.70 ± 0.22 | 84.67 ± 0.17        |
> | 0.30      | 84.97 ± 0.05 | 84.73 ± 0.05        |
> | 0.35      | 84.93 ± 0.09 | 84.80 ± 0.08        |
> | 0.40      | 84.87 ± 0.12 | 84.87 ± 0.21        |
> | 0.45      | 84.93 ± 0.12 | 84.87 ± 0.21        |
> | 0.50      | 85.10 ± 0.16 | 84.87 ± 0.21        |
>
> *Table 2: Result of the harmful data removal on ANIMAL-10 dataset.*
>
> From Table 2, we can easily find that removing harmful data identified by IF increases the model's accuracy by nearly 0.4\%, while removing harmful data identified by ClipScore may result in a decrease in the model's accuracy. This suggests that ClipScore is ineffective in identifying data that is detrimental to the model.
>
>
> *References:*
>
> [1]. Hessel, J., Holtzman, A., Forbes, M., Le Bras, R., & Choi, Y. (2021, November). CLIPScore: A Reference-free Evaluation Metric for Image Captioning. In Proceedings of the 2021 Conference on Empirical Methods in Natural Language Processing (pp. 7514-7528).

---

> ### Author Response · Authors · 2024-11-24
> **Respone to Reviewer PRsw**
>
> > **W2:**
> > The improvement is very limited. In Table 1, the accuracy is dropped but still marked in bold. It will confuse the readers. It would be better to clarify the criteria for bolding results, especially in cases where accuracy decreases
>
>
> We cannot agree your comment "The improvement is very limited". Please read ou paper carefully. Note that retraining is not a baseline, but it is the ground truth as the retraining result is obtained by removing several training pairs. Thus, there is no any method can improve it. However, retraining the model from scratch, which is the ideal case and it is definitely impossible in larger models.
>
> And this is the goal of this paper: we want to use IF as a tool for model editing without any retraining. So, a method whose performance close to the performance will be a better method. In Table 1, you can see ECIF achieves performance very close to retraining without the need for retraining, and its running time is only half that of retraining. So, these mean ECIF is very effective and efficient.
>
> Furthermore, substituting retraining is not the only contribution of this paper. Data attribution is a crucial topic that has been extensively studied in LLMs[1-3] and diffusion models[4-6], among others, but it has not been explored in MLLMs due to the complexity of multi-modal training datasets. We are the first to propose an efficient method that addresses this problem. Comprehensive experimental results demonstrate that ECIF effectively and efficiently mitigates the influence of specific samples, compared to retraining, and identifies influential data in the training set.
>
>
> *References:*
>
> [1]. Grosse, R., Bae, J., Anil, C., Elhage, N., Tamkin, A., Tajdini, A., ... & Bowman, S. R. (2023). Studying large language model generalization with influence functions. arXiv preprint arXiv:2308.03296.
>
> [2]. Choe, S. K., Ahn, H., Bae, J., Zhao, K., Kang, M., Chung, Y., ... & Xing, E. (2024). What is Your Data Worth to GPT? LLM-Scale Data Valuation with Influence Functions. arXiv preprint arXiv:2405.13954.
>
> [3]. Kwon, Y., Wu, E., Wu, K., & Zou, J. (2023). Datainf: Efficiently estimating data influence in lora-tuned llms and diffusion models. ICLR 2024.
>
> [4]. Xia, M., Malladi, S., Gururangan, S., Arora, S., & Chen, D. (2024). Less: Selecting influential data for targeted instruction tuning. ICML 2024.
>
> [5]. Dai, Z., & Gifford, D. K. (2023). Training data attribution for diffusion models. arXiv preprint arXiv:2306.02174.
>
> [6]. Zheng, X., Pang, T., Du, C., Jiang, J., & Lin, M. (2023). Intriguing properties of data attribution on diffusion models. ICLR 2024.

---

> ### Author Response · Authors · 2024-11-25
>
> Dear Reviewer PRsw,
>
> We are truly grateful for the time and effort you have dedicated to reviewing our work. We respectfully remind you that the discussion period will end in a few days. We have responded above to your concerns. We would appreciate it if you would take the time to read and comment on the responses and consider raising your score.
>
> Best, Authors

---

> > ### Author Response · Authors · 2024-11-26
> >
> > Dear Reviewer PRsw,
> >
> > Thank you again for your detailed and constructive review comments.
> >
> > We sincerely hope that the above responses can clarify any confusion regarding the comment that "the improvement is very limited."
> >
> > Should you have any questions or concerns, we would be happy to discuss and address them. We would highly appreciate your feedback on the responses.
> >
> > Best,\
> > Authors

---

> ### Author Response · Authors · 2024-11-27
>
> Dear Reviewer PRsw,
>
> Thank you again for your detailed and constructive review comments.
>
> We sincerely hope that the above responses can clarify any confusion regarding the comment that "the improvement is very limited."
>
> Should you have any questions or concerns, we would be happy to discuss and address them. We would highly appreciate your feedback on the responses.
>
> Best,
> Authors

---

> > ### Comment · Reviewer_PRsw · 2024-11-28
> >
> > Thanks for the author's response, which addresses some of my concerns but I still feel the experiments are far away from the acceptance. I adjusted my rating.
> > In addition, I appreciate the new results of Cifar100 but this data is still too limited in terms of size. I will suggest the authors consider adding the results of Imagenet. Here I agree with Reviewer 8pQR, the baselines and datasets are insufficient for empirical validation.

---

> > > ### Author Response · Authors · 2024-12-01
> > >
> > > Dear Reviewer PRsw,
> > >
> > > Here are the experiment results on **Imagenet** dataset.
> > >
> > > We evaluated the utility and efficiency of ECIF on Imagenet, and the results are shown in Table A. Random Removal refers to the practice of randomly removing data from the training dataset, followed by retraining the model or using ECIF to estimate its performance. Negative Removal refers to removing data that are mislabeled or misaligned. Positive Removal, on the other hand, refers to the removal of data that are not classified as negative.
> > >
> > > | Removal Method               | Accuracy (Mean ± Std) | Time (Mean ± Std)   |
> > > |------------------------------|-----------------------|---------------------|
> > > |Random Removal & Retrain | 64.02+-0.18| 560.75+-4.31|
> > > | Random Removal & IF Update   | 63.67 ± 0.25         | 127.03 ± 1.77       |
> > > |Positive Removal & Retrain | 63.83+-1.23 | 716.20+-12.00|
> > > | Positive Removal & IF Update | 63.73 ± 0.21         | 122.98 ± 5.81       |
> > > | Negative Removal & Retrain   | 63.57 ± 0.09         | 291.69 ± 0.19       |
> > > | Negative Removal & IF Update | 63.80 ± 0.08         | 147.30 ± 8.21       |
> > >
> > > *Table A: Performance comparison of retraining and ECIF on Imagenet*
> > >
> > > The results in Table 1 demonstrate that across all three removal scenarios, the accuracy of the model, as estimated by ECIF, is comparable to that of retraining.
> > >
> > > In all the three scenarios, the ECIF method is significantly faster than retraining, and the runtime difference is significantly greater in the positive removal scenario, reaching only 1/6 of the runtime required for retraining.
> > >
> > > The presented results demonstrate the superior accuracy and robustness of our ECIF method on ImageNet. We respectfully request your evaluation of these results and hope you will consider raising your score. If you have any questions or concerns, we would be glad to discuss and address them. Your feedback on our responses would be greatly appreciated.
> > >
> > > Best,\
> > > Authors

---

> ### Author Response · Authors · 2024-11-30
>
> Dear Reviewer PRsw,
>
> We hope you had a wonderful Thanksgiving. Thank you very much for taking the time to review our responses and reassess our work. We sincerely appreciate your acknowledgment of our response.
>
> We are conducting experiments on ImageNet, which are still on-going due to time constraints. We have employed ClipScore as a baseline, and the results are listed above.
>
> For this data attribution task, there are currently a limited number of suitable baselines. The one we have found most suitable so far is ClipScore, and we have conducted relavent experiments.
>
> Thank you once again for your contribution to ICLR! Wishing you all the best with your current and future submissions.
>
> Sincerely,\
> Authors

---

> ### Author Response · Authors · 2024-12-02
>
> Dear Reviewer PRsw,
>
> I hope this message finds you well. I am writing to follow up, as today marks the deadline for providing feedback on our rebuttal discussions. Your insights are invaluable in addressing any remaining concerns and refining our submission to the best possible version.
>
> We have conducted additional experiments on ImageNet and would greatly appreciate your review and feedback on the results. If any aspects of the rebuttal require further clarification or discussion, please let us know, and we will respond promptly. If your concerns have already been addressed, we kindly ask that you consider reflecting this in your review score.
>
> Thank you for your time and effort during this review process. We sincerely appreciate your valuable feedback and look forward to hearing from you.
>
> Best,\
> Authors

---

### Official Review · Reviewer_DHqx · 2024-11-02

**Soundness:** 3
**Presentation:** 3
**Contribution:** 3
**Rating:** 6
**Confidence:** 3

**Summary:**

This paper proposes the Extended Influence Function for Contrastive Loss (ECIF), a method designed to improve data valuation in multimodal large language models (MLLMs). ECIF addresses the limitations of traditional influence functions by accounting for the roles of both positive and negative samples in contrastive learning, enabling a more accurate assessment of data impact and model alignment.

**Strengths:**

1. ECIF is a novel extension of influence functions tailored specifically for contrastive loss, allowing it to consider both positive and negative samples' impacts, which is essential for contrastive learning in MLLMs.
2. By providing a closed-form approximation, ECIF avoids the need for retraining, making it practical and computationally efficient for large models.
3. The paper demonstrates ECIF's effectiveness across multiple tasks, including data valuation, misalignment detection, and misprediction trace-back, showcasing its versatility.

**Weaknesses:**

1. ECIF assumes a linear combination of influences from positive and negative samples, which may not accurately reflect real-world complexities in multimodal contrastive tasks.
2. The experiments focus on relatively clean and well-defined datasets, limiting the assessment of ECIF's robustness in more challenging, noisy, or open-domain multimodal datasets.

**Questions:**

Please see the weakness part.

---

> ### Author Response · Authors · 2024-11-24
> **Respone to Reviewer DHqx**
>
> ###  Weaknesses:
>
> >**W1:**
> >ECIF assumes a linear combination of influences from positive and negative samples, which may not accurately reflect real-world complexities in multimodal contrastive tasks.
>
> Note that we **do not assume** the influnce is a combination from positive and negative samples, it is derived from mathematical proof given in Property 5.1.. Our proof is based on the basic **definition** of IF in robust statistics.
>  We separate into positive and negative samples is to give better illustration.
>
>
> In IF, each specific data point corresponds to a single term in the loss function. As a result, the influence of different data points is linear with respect to the loss function. Up-weighting individual data points separately and summing their influence functions is equivalent to up-weighting them simultaneously and deriving a combined IF. Therefore, the IF for multiple data points is actually the linear combination of the IF for individual data.
>
> A similar structure applies to contrastive loss. The positive and negative influences of a single data point correspond to two independent terms in the contrastive loss function. In Property 5.1, we follow the definition of IF and linearly combined the positive-IF and negative-IF as the IF for one data.
>
> >**W2:**
> >The experiments focus on relatively clean and well-defined datasets, limiting the assessment of ECIF's robustness in more challenging, noisy, or open-domain multimodal datasets.
>
> In response to concerns about the robustness of ECIF on challenging, noisy, or open-domain multimodal datasets, we have conducted experiments using synthetic noisy Imagenetee and Cifar 10. Our method showed excellent performance in these scenarios.
>
> Our method concentrates on the alignment ability of CLIP, a multimodal encoder integral to MLLMs. CLIP is pre-trained on extensive text-image paired datasets, which enhances its ability to align two modalities effectively. This pre-trained model is then incorporated into MLLMs, which are subsequently trained on conversational datasets. Subsequently, the alignment capabilities of MLLMs are largely inherited from CLIP, which was trained on non-conversational datasets. Therefore, our experiments can focus on paired datasets without necessarily requiring open-domain multimodal datasets.
>
>
> To further demonstrate the ability of our ECIF, we conduct harmful removal experiments on real-world noisy dataset ANIMAL-10. Here we use CLIPScore[3], a basic data evaluation method for MLLM as another reference method. This method is model-independent and is limited to evaluating data quality rather than assessing the contribution of the data to the model.
>
> The harmful data identified by CLIPScore correspond to those with low scores. We first remove the harmful data recognized by IF and ClipScore, then retrain the model and test the performance. The results are showed in Table 1.
>
> | Threshold | Accuracy(IF)  | Accuracy(ClipScore) |
> | --------- | ------------ | ------------------- |
> | 0.05      | 84.73 ± 0.21 | 84.90 ± 0.00        |
> | 0.10      | 84.70 ± 0.24 | 84.90 ± 0.00        |
> | 0.15      | 84.80 ± 0.29 | 84.90 ± 0.00        |
> | 0.20      | 84.67 ± 0.09 | 83.57 ± 0.05        |
> | 0.25      | 84.70 ± 0.22 | 84.67 ± 0.17        |
> | 0.30      | 84.97 ± 0.05 | 84.73 ± 0.05        |
> | 0.35      | 84.93 ± 0.09 | 84.80 ± 0.08        |
> | 0.40      | 84.87 ± 0.12 | 84.87 ± 0.21        |
> | 0.45      | 84.93 ± 0.12 | 84.87 ± 0.21        |
> | 0.50      | 85.10 ± 0.16 | 84.87 ± 0.21        |
>
> *Table 1: Result of the harmful data removal on ANIMAL-10 dataset.*
>
> From Table 1, it is evident that removing harmful data identified by ECIF results in a significant improvement in the model's accuracy, increasing it by nearly 0.4%. In contrast, removing harmful data identified by ClipScore may lead to a decrease in accuracy, highlighting the ineffectiveness of ClipScore in identifying detrimental data. These findings underscore the robustness and reliability of our method ECIF in accurately identifying and addressing harmful data to enhance model performance.
>
> *References:*
>
> [1]. Wang, S., Tan, Z., Guo, R., & Li, J. (2023, December). Noise-Robust Fine-Tuning of Pretrained Language Models via External Guidance. In Findings of the Association for Computational Linguistics: EMNLP 2023 (pp. 12528-12540).
>
> [2]. Feng, C., Tzimiropoulos, G., & Patras, I. (2024, October). CLIPCleaner: Cleaning Noisy Labels with CLIP. In Proceedings of the 32nd ACM International Conference on Multimedia (pp. 876-885).
>
> [3]. Hessel, J., Holtzman, A., Forbes, M., Le Bras, R., & Choi, Y. (2021, November). CLIPScore: A Reference-free Evaluation Metric for Image Captioning. In Proceedings of the 2021 Conference on Empirical Methods in Natural Language Processing (pp. 7514-7528).

---

> ### Author Response · Authors · 2024-11-25
>
> Dear Reviewer DHqx,
>
> We are truly grateful for the time and effort you have dedicated to reviewing our work. We respectfully remind you that the discussion period will end in a few days. We have responded above to your concerns. We would appreciate it if you would take the time to read and comment on the responses and consider raising your score.
>
> Best,\
> Authors

---

> > ### Author Response · Authors · 2024-11-26
> >
> > Dear Reviewer DHqx,
> >
> > Thank you again for your detailed and constructive review comments. Should you have any questions or concerns, we would be happy to discuss and address them. We would highly appreciate your feedback on the responses.
> >
> >
> >
> > Best, \
> > Authors

---

> ### Author Response · Authors · 2024-11-27
>
> Dear Reviewer DHqx,
>
> Thank you again for your detailed and constructive review comments. Should you have any questions or concerns, we would be happy to discuss and address them. We would highly appreciate your feedback on the responses.
>
> Best,
> Authors

---

> ### Author Response · Authors · 2024-11-30
>
> Dear Reviewer DHqx,
>
> We hope you had a wonderful Thanksgiving. Once again, we sincerely thank you for your valuable time and effort, and we fully understand your busy schedule. We noticed that we haven’t received any feedback from you yet. In the meantime, we have addressed the concerns of reviewer PRsw, who has expressed a positive attitude and increased the score.
>
> We would be truly grateful if you could let us know whether there are any additional questions or concerns we can address before the rebuttal deadline. Your feedback is invaluable and plays a crucial role in strengthening our work.
>
> Thank you once again for your contribution to ICLR! Wishing you all the best with your current and future submissions.
>
> Sincerely,\
> The Authors

---

> ### Author Response · Authors · 2024-12-01
>
> Dear Reviewer DHqx,
>
> We sincerely appreciate your valuable contributions to ICLR. As the extended discussion period will conclude in a few days, we kindly remind you to share any final input.
>
> Having addressed the concerns you raised, we respectfully request that you review our clarifications and consider providing further comments or updating your score. We trust that you will re-evaluate our paper based on the additional details provided.
>
> Thank you once again for your thoughtful contributions. We look forward to receiving your final decision.
>
> Best regards,\
> Authors

---

> ### Author Response · Authors · 2024-12-02
>
> Dear Reviewer DHqx,
>
> I hope this message finds you well. I am writing to kindly follow up, as today marks the final day to provide feedback on our rebuttal discussions. Your insights are invaluable in addressing any remaining concerns and refining our submission to its best possible version.
>
> If there are any aspects of the rebuttal that require further clarification or discussion, please do not hesitate to let us know. We will address them promptly. If your concerns have already been addressed, we kindly ask that you consider reflecting this in your review score.
>
> Thank you for your time and effort during this review process. We sincerely appreciate your valuable feedback and look forward to hearing from you.
>
> Best regards,\
> Authors

---

### Official Review · Reviewer_9qPR · 2024-11-02

**Soundness:** 2
**Presentation:** 2
**Contribution:** 2
**Rating:** 5
**Confidence:** 2

**Summary:**

The paper introduces the Extended Influence Function for Contrastive Loss (ECIF) to address data misalignment in multimodal large language models (MLLMs). ECIF is designed to efficiently evaluate data influence for contrastive learning, considering both positive and negative samples. It provides a closed-form approximation, avoiding retraining, and improves the transparency and interpretability of MLLMs by accurately assessing data impact and alignment.

**Strengths:**

- **Dual-Perspective Data Valuation**: The proposed ECIF is the first dual-perspective data valuation method for multimodal large language models.

- **Applicable to Various Tasks**: The authors developed algorithms for different tasks, including identifying the most valuable data for specific tasks, detecting data misalignment, and tracing mispredictions.

- **Efficient Sample Influence Removal**: Experimental results show that ECIF is more efficient than retraining at removing the influence of samples and identifying influential data in the training set.

**Weaknesses:**

- **Questionable Design Rationale**: The authors applied the Influence Function to contrastive learning, but the overall design's rationale is debatable.

- **Insufficient Experiments**: The experiments are limited, with too few baselines and datasets, leading to insufficient empirical validation.

- **Lack of Clear Explanation**: Despite the use of many formulas, the paper lacks a clear explanation of why the Influence Function was used.

**Questions:**

- The number of datasets and baselines used in the experimental section is limited, which fails to sufficiently demonstrate the generalizability and effectiveness of the proposed method.

- What is the necessity of using the Influence Function?

---

> ### Author Response · Authors · 2024-11-24
> **Response to Reviewer 9qPR**
>
> ### Weaknesses:
> >**W1:**
> >Questionable Design Rationale: The authors applied the Influence Function to contrastive learning, but the overall design's rationale is debatable.
>
> Training data attribution is one of the most important topics in large models, and there are many papers on it, such as [1-3]. IF is a basic and classic method for training data attribution. Thus, we do not agree with your point. We would appreciate it if you could provide a more detailed explanation of why the overall design's rationale is considered debatable.
>
> >**W2:**
> >Insufficient Experiments: The experiments are limited, with too few baselines and datasets, leading to insufficient empirical validation.
>
> We do not agree. As you can see from the experiments, we have evaluated the performance of ECIF in analyzing data influence at the parameter level (see Table 1 in the paper). Subsequently, we use ECIF to identify the most valuable and harmful data for a specific task, as illustrated in Figure 1. Additionally, we use ECIF to identify training data that are most relevant to specific mispredicted test samples. We also detect misaligned data pairs within the training dataset. Comprehensive experiments have strongly demonstrated the effectiveness of our method.
>
> So, please give more details on the limited experiments you mentioned.
>
> Because the attribution of the training data, specifically in the context of MLLM, has not been extensively studied, suitable baselines for this task are lacking. As we mentioned, this is the first paper studying data influence for contrastive learning, and there are no commonly-sued baselines.
>
> To provide a method as the reference, we adopt CLIPScore[4], a basic data evaluation method, as the baseline for MLLM. This method is model-independent and is limited to evaluating data quality rather than assessing the contribution of the data to the model.
> We use the CLIPScore to conduct harmful removal experiments.
>
> The harmful data identified by CLIPScore correspond to those with low scores. We retrain the model after removing these data and test the model performance.
>
> Here  we conduct  experiments on a new dataset: ANIMAL-10 dataset. We first remove the harmful data recognized by IF and ClipScore, then retrain the model and test the performance. The results are showed in Table B.
>
> | Threshold | Accuracy(IF)  | Accuracy(ClipScore) |
> | --------- | ------------ | ------------------- |
> | 0.05      | 84.73 ± 0.21 | 84.90 ± 0.00        |
> | 0.10      | 84.70 ± 0.24 | 84.90 ± 0.00        |
> | 0.15      | 84.80 ± 0.29 | 84.90 ± 0.00        |
> | 0.20      | 84.67 ± 0.09 | 83.57 ± 0.05        |
> | 0.25      | 84.70 ± 0.22 | 84.67 ± 0.17        |
> | 0.30      | 84.97 ± 0.05 | 84.73 ± 0.05        |
> | 0.35      | 84.93 ± 0.09 | 84.80 ± 0.08        |
> | 0.40      | 84.87 ± 0.12 | 84.87 ± 0.21        |
> | 0.45      | 84.93 ± 0.12 | 84.87 ± 0.21        |
> | 0.50      | 85.10 ± 0.16 | 84.87 ± 0.21        |
>
> *Table B: Result of the harmful data removal on ANIMAL-10 dataset.*
>
> From Table B, we can easily find that removing harmful data identified by IF increases the model's accuracy by nearly 0.4\%, while removing harmful data identified by ClipScore may result in a decrease in the model's accuracy. This suggests that ClipScore is ineffective in identifying data that is detrimental to the model.
>
> *References:*
>
> [1]. Grosse, R., Bae, J., Anil, C., Elhage, N., Tamkin, A., Tajdini, A., ... & Bowman, S. R. (2023). Studying large language model generalization with influence functions. arXiv preprint arXiv:2308.03296.
>
> [2]. Choe, S. K., Ahn, H., Bae, J., Zhao, K., Kang, M., Chung, Y., ... & Xing, E. (2024). What is Your Data Worth to GPT? LLM-Scale Data Valuation with Influence Functions. arXiv preprint arXiv:2405.13954.
>
> [3]. Kwon, Y., Wu, E., Wu, K., & Zou, J. (2023). Datainf: Efficiently estimating data influence in lora-tuned llms and diffusion models. arXiv preprint arXiv:2310.00902.
>
> [4]. Hessel, J., Holtzman, A., Forbes, M., Le Bras, R., & Choi, Y. (2021, November). CLIPScore: A Reference-free Evaluation Metric for Image Captioning. In Proceedings of the 2021 Conference on Empirical Methods in Natural Language Processing (pp. 7514-7528).

---

> > ### Author Response · Authors · 2024-11-30
> >
> > Dear Reviewer 9qPR,
> >
> > We hope you had a wonderful Thanksgiving. Once again, we sincerely thank you for your valuable time and effort, and we fully understand your busy schedule. We noticed that we haven’t received any feedback from you yet. In the meantime, we have addressed the concerns of reviewer PRsw, who has expressed a positive attitude and increased the score.
> >
> > We would be truly grateful if you could let us know whether there are any additional questions or concerns we can address before the rebuttal deadline. Your feedback is invaluable and plays a crucial role in strengthening our work.
> >
> > Thank you once again for your contribution to ICLR! Wishing you all the best with your current and future submissions.
> >
> > Sincerely,\
> > The Authors

---

> ### Author Response · Authors · 2024-11-24
> **Respone to Reviewer 9qPR**
>
> ### Weaknesses
> >**W3:**
> >Lack of Clear Explanation: Despite the use of many formulas, the paper lacks a clear explanation of why the Influence Function was used.
>
>
> We have included details on why we use IF at the beginning of the paper. Please read our paper carefully.
>
> There are two primary data attribution methods:
>
> The first is **Shapley Value**, based on **sampling**. This method, while theoretically sound, requires multiple retrainings of the model, making it computationally expensive and impractical for large-scale datasets and models due to the need to evaluate all possible subsets of data.
>
> The second method, Influence Function, is generally more scalable for large datasets and models compared to sampling-based methods. To evaluate a specific data point's contribution to model performance, one could remove the data point, retrain the model, and then test its performance. A significant drop in performance would indicate that the data point is crucial for model training. Influence Functions efficiently estimate data contribution without the need for model retraining. This method is efficient because, firstly, Influence Functions do not require exhaustive sampling of all possible data point combinations. Secondly, Influence Functions can estimate the retrained model after data removal.
>
> Furthermore, Influence Functions have been successfully applied in LLMs, as demonstrated in [1-4], but have not yet been applied to MLLMs, which is the focus of this paper.
>
> *References:*
>
> [1]. Grosse, R., Bae, J., Anil, C., Elhage, N., Tamkin, A., Tajdini, A., ... & Bowman, S. R. (2023). Studying large language model generalization with influence functions. arXiv preprint arXiv:2308.03296.
>
> [2]. Choe, S. K., Ahn, H., Bae, J., Zhao, K., Kang, M., Chung, Y., ... & Xing, E. (2024). What is Your Data Worth to GPT? LLM-Scale Data Valuation with Influence Functions. arXiv preprint arXiv:2405.13954.
>
> [3]. Kwon, Y., Wu, E., Wu, K., & Zou, J. (2023). Datainf: Efficiently estimating data influence in lora-tuned llms and diffusion models. ICLR 2024.
>
> [4]. Xia, M., Malladi, S., Gururangan, S., Arora, S., & Chen, D. (2024). Less: Selecting influential data for targeted instruction tuning. ICML 2024.

---

> ### Author Response · Authors · 2024-11-24
> **Response to Reviewer 9qPR**
>
> ### Questions:
> >**Q1:**
> >The number of datasets and baselines used in the experimental section is limited, which fails to sufficiently demonstrate the generalizability and effectiveness of the proposed method.
>
>
> We extened our method to **Cifar100**, and the efficiency and utility results are shown here:
>
>
> | Method                     | Accuracy (%) | Time (s)     |
> | -------------------------- | ------------ | ------------ |
> | Random Removal & Retrain   | 73.50 ± 0.35 | 12.56 ± 0.37 |
> | Random Removal & IF Update | 73.00 ± 0.20 | 7.40 ± 0.11  |
>
> *Table 1: Result of the efficiency and utility experiment on Cifar 100 dataset.*
>
> From Table 1, we can find that ECIF achieves a performance comparable to that of retraining, while requiring only half the running time. This indicatesd that our method is still effective for complex large dataset.
>
>
> Given the limited research on training data attribution within the context of MLLM, suitable baselines for this task are scarce. Therefore, we utilize CLIPScore[1] as a baseline for our task mislabeled trace back and misaligned detection tasks. However, this method is limited to assessing data quality rather than evaluating the contribution of data to model performance.
>
>
> We conduct harmful removal experiments on ANIMAL-10 dataset, a real world noisy dataset. The harmful data identified by CLIPScore correspond to those with low scores. We first remove the harmful data recognized by IF and ClipScore, then retrain the model and test the performance. The results are showed in Table A.
>
> | Threshold | Accuracy(IF)  | Accuracy(ClipScore) |
> | --------- | ------------ | ------------------- |
> | 0.05      | 84.73 ± 0.21 | 84.90 ± 0.00        |
> | 0.10      | 84.70 ± 0.24 | 84.90 ± 0.00        |
> | 0.15      | 84.80 ± 0.29 | 84.90 ± 0.00        |
> | 0.20      | 84.67 ± 0.09 | 83.57 ± 0.05        |
> | 0.25      | 84.70 ± 0.22 | 84.67 ± 0.17        |
> | 0.30      | 84.97 ± 0.05 | 84.73 ± 0.05        |
> | 0.35      | 84.93 ± 0.09 | 84.80 ± 0.08        |
> | 0.40      | 84.87 ± 0.12 | 84.87 ± 0.21        |
> | 0.45      | 84.93 ± 0.12 | 84.87 ± 0.21        |
> | 0.50      | 85.10 ± 0.16 | 84.87 ± 0.21        |
>
> *Table A: Result of the harmful data removal on ANIMAL-10 dataset.*
>
> From Table A, we can easily find that removing harmful data identified by IF increases the model's accuracy by nearly 0.4\%, while removing harmful data identified by ClipScore may result in a decrease in the model's accuracy. This suggests that ClipScore is ineffective in identifying data that is detrimental to the model.
>
>
> >**Q2:**
> >What is the necessity of using the Influence Function?
>
>
> In this paper, we propose an efficient data attribution method to tackle the issue of misalignment in the training datasets of MLLM. And for data attribution, there are two predominant data attribution techniques:
>
> **Shapley Value**: This method relies on sampling and necessitates numerous model retrainings. This requirement makes it computationally expensive and impractical for large-scale datasets and models, as it involves evaluating all possible subsets of data.
>
> **Influence Functions**: To assess how a specific data point within a training dataset impacts model performance, one approach is to remove the data point, retrain the model, and then evaluate its performance. A notable decrease in performance would indicate that the data point plays a crucial role in the model's effectiveness. And Influence Functions offer an efficient alternative to estimate this contribution without the need for retraining. This method can estimate the model's performance after data removal, thereby avoiding any need for retraining.
>
>
> Moreover, while Influence Functions have been successfully applied in Large Language Models (LLMs), as shown in [1-4], they have not yet been utilized for MLLMs, which is the main focus of this paper.
>
> *References*
>
> [1]. Hessel, J., Holtzman, A., Forbes, M., Le Bras, R., & Choi, Y. (2021, November). CLIPScore: A Reference-free Evaluation Metric for Image Captioning. In Proceedings of the 2021 Conference on Empirical Methods in Natural Language Processing (pp. 7514-7528).
>
> [2]. Grosse, R., Bae, J., Anil, C., Elhage, N., Tamkin, A., Tajdini, A., ... & Bowman, S. R. (2023). Studying large language model generalization with influence functions. arXiv preprint arXiv:2308.03296.
>
> [3]. Choe, S. K., Ahn, H., Bae, J., Zhao, K., Kang, M., Chung, Y., ... & Xing, E. (2024). What is Your Data Worth to GPT? LLM-Scale Data Valuation with Influence Functions. arXiv preprint arXiv:2405.13954.
>
> [4]. Kwon, Y., Wu, E., Wu, K., & Zou, J. (2023). Datainf: Efficiently estimating data influence in lora-tuned llms and diffusion models. ICLR 2024.
>
> [5]. Xia, M., Malladi, S., Gururangan, S., Arora, S., & Chen, D. (2024). Less: Selecting influential data for targeted instruction tuning. ICML 2024.

---

> ### Author Response · Authors · 2024-11-25
>
> Dear Reviewer 9qPR,
>
> We are truly grateful for the time and effort you have dedicated to reviewing our work. We respectfully remind you that the discussion period will end in a few days. We have responded above to your concerns. We would appreciate it if you would take the time to read and comment on the responses and consider raising your score.
>
> Best,
> Authors

---

> ### Author Response · Authors · 2024-11-26
>
> Dear Reviewer 9qPR,
>
> Thank you again for your detailed and constructive review comments. Should you have any questions or concerns, we would be happy to discuss and address them. We would highly appreciate your feedback on the responses.
>
>
>
> Best, \
> Authors

---

> ### Author Response · Authors · 2024-11-27
>
> Dear Reviewer 9qPR,
>
> Thank you again for your detailed and constructive review comments. Should you have any questions or concerns, we would be happy to discuss and address them. We would highly appreciate your feedback on the responses.
>
> Best,
> Authors

---

> ### Author Response · Authors · 2024-12-01
>
> Dear Reviewer 9qPR,
>
> Thank you for your valuable contributions to ICLR. We kindly remind you that the extended discussion period will end in a few days.
>
> Since we have addressed the concerns you raised, we kindly request that you review our clarifications and consider providing additional comments or updating your score. We trust that you, as the reviewer, will re-evaluate our paper based on our clarifications.
>
> Thank you for your valuable contributions, and we kindly await your thoughtful final decision.
>
> Best,\
> Authors

---

> ### Author Response · Authors · 2024-12-02
>
> Dear Reviewer 9qPR,
>
> I hope this message finds you well. I am writing to follow up as today marks the final day for providing feedback on our rebuttal discussions. Your insights are invaluable in addressing any remaining concerns and ensuring the best possible version of our submission.
>
> If there are any points in the rebuttal that require further clarification or discussion, I would be more than happy to address them promptly. I truly appreciate the time and effort you dedicate to this review process. Otherwise, we respectfully request you to consider raising your rating score accordingly if your concerns are alleviated.
>
> Thank you very much for your consideration, and I look forward to your feedback.
>
> Best regards,\
> Authors

---

### Author Response · Authors · 2024-11-24
**General Response (Revision Updates)**

Dear Reviewers and ACs,

Thank you for providing your detailed, insightful, and helpful review comments. We addressed your concerns in our responses to each reviewer and revised the paper based on your suggestions. We greatly appreciate how the reviewers' comments have strengthened our paper. Below, we summarize the key revisions made.

1. We add experiments on Cifar-100 dataset. The results are listed in Appendix F.2 in the revision.
2. We add experiments on ANIMAL-10 dataset, a **real-world noisy dataset**, to further demonstrate the robustness of our ECIF. The results are listed in Appendix F.3 in the revision.
3. We use ClipScore as a **baseline** method and conduct harmful data removal, while ClipScore cannot attribute data but only evaluate data quality. This is due to the lack of alternative baseline methods available for this task. The results are also listed in Appendix F.3 in the revision.
4. We update the description of retraining from 'baseline' to 'ground truth' in the Baselines and Evaluation Metric part.

We noticed that reviewers may have some misunderstandings when comparing retrain and ECIF. For our data attribution task, retrain serves as a ground truth rather than a baseline. By retraining the model after removing the data to be evaluated, we quantitatively provide the influence score of the data. However, retraining is impossible for large models and thus it is the ideal cas rather than an efficient method.

Our proposed ECIF method is an efficient and accurate approximation of this ground truth. As shown in Table 1, the estimation results of ECIF are very similar to those of the retrain method, while the runtime is significantly reduced. These results show the efficiency of our method. Therefore, to avoid misinterpretation, we update the description of retraining from 'baseline' to 'ground truth' in the Baselines and Evaluation Metric part.


We sincerely hope that the responses and the revised paper assist in your re-evaluation of our paper.

Sincerely,\
Authors

---

### Meta-Review · Area_Chair_4DSE · 2024-12-15

**Metareview:**

This paper proposes the Extended Influence Function for Contrastive Loss (ECIF), designed to evaluate data influence in contrastive learning models for multimodal large language models (MLLMs). While the idea of using influence functions for contrastive loss is novel, the paper suffers from significant shortcomings. The experiments are limited in scope, relying on small and simplistic datasets that do not adequately demonstrate the robustness or scalability of the proposed method. Additionally, key assumptions, such as the linearity of influence from positive and negative samples, lack sufficient justification. Despite attempts to address these issues during the rebuttal, the revisions were not comprehensive enough to resolve major concerns. Therefore, this paper is not ready for acceptance in its current form.

**Additional Comments On Reviewer Discussion:**

The reviewers raised concerns regarding the experimental design, dataset diversity, and theoretical assumptions underpinning ECIF. While the authors provided additional experiments and clarified some points, these efforts fell short of addressing the broader issues. Specifically, the reliance on limited datasets like CIFAR-100 and ANIMAL-10 restricts the generalizability of the results, and the scalability to real-world scenarios remains untested. The lack of thorough comparisons with stronger baselines further undermines the validity of the conclusions. These unresolved concerns ultimately justify the decision to reject the paper.

---

### Decision · Program_Chairs · 2025-01-22

Reject